# Altered respiratory virome and serum cytokine profile associated with recurrent respiratory tract infections in children

Yanpeng Li[1,2,8], Xuemin Fu[2,8], Jinmin Ma [3,8], Jianhui Zhang[1,8], Yihong Hu[2], Wei Dong[4], Zhenzhou Wan[5], Qiongfang Li[3], Yi-Qun Kuang[6], Ke Lan[2,7], Xia Jin[2], Jian-Hua Wang[2] & Chiyu Zhang [1,2]

Recurrent acute respiratory tract infections (ARTIs) affect a large population, yet the specific decisive factors are largely unknown. Here we study a population of 4407 children diagnosed with ARTI, comparing respiratory virome and serum cytokine profiles associated with multiple ARTIs and single ARTI during a six-year period. The relative abundance of *Propionibacterium* phages is significantly elevated in multiple ARTIs compared to single ARTI group. Serum levels of TIMP-1 and PDGF-BB are markedly increased in multiple ARTIs compared to single-ARTI and non-ARTI controls, making these two cytokines potential predictors for multiple ARTIs. The presence of *Propionibacterium* phages is associated with higher levels of TIMP-1 and PDGF-BB. Receiver operating characteristic (ROC) curve analyses show that the combination of TIMP-1, PDGF-BB and *Propionibacterium* phages could be a strong predictor for multiple ARTIs. These findings indicate that respiratory microbe homeostasis and specific cytokines are associated with the onset of multiple ARTIs over time.

[1] The Joint Center for Infection and Immunity between Guangzhou Institute of Pediatrics, Guangzhou Women and Children's Medical Center (Guangzhou, 510623, China) and Institut Pasteur of Shanghai, Chinese Academy of Sciences (CAS), Shanghai 200031, China. [2] Key Laboratory of Molecular Virology & Immunology, Institut Pasteur of Shanghai, Chinese Academy of Sciences, Shanghai 200031, China. [3] BGI-Shenzhen, Shenzhen 518083, China. [4] Pediatric Department, Shanghai Nanxiang Hospital, Jiading District, Shanghai 201800, China. [5] Medical Laboratory of Taizhou Fourth People's Hospital, Taizhou, Jiangsu 225300, China. [6] Institute of Infection and Immunity, Henan University, Kaifeng 475000, China. [7] Present address: State Key Laboratory of Virology, College of Life Sciences, Wuhan University, Wuhan 430072, China. [8] These authors contributed equally: Yanpeng Li, Xuemin Fu, Jinmin Ma, Jianhui Zhang. Correspondence and requests for materials should be addressed to J.-H.W. (email: jh_wang@ips.ac.cn) or to C.Z. (email: zhangcy1999@ips.ac.cn)

Acute respiratory tract infections (ARTIs) are the most common cause of morbidity and mortality among children under the age of five worldwide, with 4 million annual deaths estimated[1,2]. ARTIs can be classified as upper respiratory tract infections (URIs) of which most are caused by viral infections, or lower respiratory tract infections (LRIs) which are often associated with both bacterial and viral pathogens. These illnesses frequently occur during the first two years of life when a child's immune system is not fully developed[3]. As expected, recurrent ARTIs decline progressively in frequency with age increase[4,5]. Some children rarely develop symptomatic ARTIs, whereas others frequently experience multiple ARTIs over time during childhood. Frequent recurrence of ARTIs poses a considerable psychological and physical stress on children and an economic burden on parents. To ascertain specific host factors and respiratory pathogens that might influence the susceptibility of a child to frequent (multiple) ARTIs may help to develop early diagnosis tools and early intervention strategies that enable reduction of morbidity and mortality caused by these diseases.

Infections and immune status are thought to be the main determinants of ARTIs in children[6], yet the specific pathogen or host factors that might contribute to the onset of multiple ARTIs are largely unknown. A few recent studies investigate viral epidemiology longitudinally in recurrent ARTIs in children using a multiplex RT-PCR panel for respiratory virus detection[4,5]. No specific respiratory viruses are found to be responsible for the occurrence of multiple ARTIs. Given the current multiplex RT-PCR panel has limited capacity to cover a large number of pathogens, it is generally agreed that the next generation sequencing (NGS) would be a better tool for an unbiased detection of the entire population of respiratory viruses. Indeed, some cross-sectional studies of the respiratory virome reveal great viral diversity among children and adults with ARTIs using pooled samples[7–10]. As yet, there is no longitudinal data available on the association between respiratory virome dynamics and recurrent ARTIs among children.

Among host factors, cytokines are known to play important antiviral roles, but they may also cause damage to respiratory tract and lung[11,12]. Therefore, the balance between pro-inflammatory and anti-inflammatory cytokines determines the outcome of an immune response to an infection. It is well established that persistent chronic inflammation will damage the development of airway and lung tissues, leading to subsequent respiratory diseases such as asthma[11,13–15]. To date, only a limited number of studies investigate cytokine profiles in children with ARTIs, mostly in relation to specific respiratory virus infection (e.g., influenza, rhinovirus)[11,12,16]. No studies are designed to measure cytokine profiles longitudinally among children with multiple ARTIs. In this study, we characterized the longitudinal changes in respiratory virome and cytokine profile in clinical samples obtained from children with either multiple ARTIs or single ARTI using metagenomic sequencing and proteomic chip-based cytokine assay, and discovered a number of factors that are significantly associated with the susceptibility of frequent ARTIs.

## Results

**Respiratory virome of single ARTI and multiple ARTIs.** To examine which respiratory pathogens and/or cytokines are associated with multiple ARTIs in children, we first established a cohort of 4407 children who experienced ARTIs during a seven-year period between 2009–2015 in Shanghai, China, and collected nasopharyngeal swabs and serum samples longitudinally (Fig. 1). From those subjects with sufficient swab and serum samples and unequivocal clinical information, we first characterized the

respiratory virome using metageonomic sequencing in 61 children who experienced two or more episodes of ARTIs, and 48 control children who experienced one episode (Fig. 1). In the 61 children with multiple ARTIs over time, there are 29, 23, 8, and 1 with two, three, four, and five episodes of infections, respectively. A single-ARTI group was selected to match the multiple ARTIs group with respect to age, gender, clinical presentations and laboratory tests (Table 1 and Supplementary Table 1).

The respiratory virome analyses revealed the existence of 34 common respiratory viruses, 39 anelloviruses, and 4 main bacteriophage families (Supplementary Fig. 1). There were significantly higher rates of detection (79.9%) and co-detection (41.5%) of common respiratory viruses in the multiple-ARTIs children than the single-ARTI group (56.3% and 22.9%, respectively) (Fig. 2a). These results confirmed that respiratory viruses are ubiquitously present in recurrent ARTIs. In accord with the idea of multiple respiratory virus airway infections overtime, virome analyses revealed a significantly higher Shannon diversity and Chao richness of respiratory viruses in the multiple-ARTIs children than the single-ARTI ones ($P < 0.01$), and at almost every episode ($P < 0.05$) (Fig. 2b, c, nonparametric Kruskal-Wallis Test).

In view of the fact that respiratory viruses identified in this study contained 22 pathogenic respiratory viruses causing clinical or subclinical infections and 12 persistent viruses rarely involved in ARTIs (e.g., human herpesviruses, polyomaviruses, adeno-associated viruses, and papillomaviruses) (Supplementary Fig. 1), data were re-analyzed by excluding those persistent viruses. A consistent trend in the detection/co-detection rates, as well as Shannon diversity index and Chao richness score was observed in the single-ARTI and multiple-ARTIs groups (Fig. 2d–f).

However, the same was not observed for anelloviruses that do not cause ARTIs, but are often detected in healthy individuals and patients with various diseases[17–20]. For the 39 different anelloviruses, the detection and the co-detection rates were similar in the multiple-ARTIs and single-ARTI children, and both rates remained stable over time (Fig. 2g). Consistently, the Shannon diversity and Chao richness of anellovirus were not significantly different between the two patient groups (Fig. 2h, i). These results are in agreement with the general knowledge that anelloviruses are not pathogenic in the respiratory tract.

Despite most ARTIs are viral origin, bacterial infections are the second most common cause of respiratory diseases. Because of failure in deciphering the respiratory microbiota directly, we examined bacteriophages that co-evolve with their host bacteria and reflect indirectly the existence of bacteria. There was no significant difference in the overall detection rate of bacteriophages between the multiple-ARTIs and single-ARTI children (Supplementary Fig. 2a). The Shannon diversity and Chao richness of bacteriophages had consistent levels and were not significantly different between the two patient groups (Supplementary Fig. 2b and c).

Results showed that human pathogens rhinovirus (HRV) (24.1%), influenza virus (IFV) (19.8%) and enterovirus (HEV) (15.6%) were the most abundant respiratory viruses detected by NGS (Fig. 3a); among anelloviruses, the top three were torque teno virus (TTV) (42.5%), TTV-like mini virus (TTMV) (32.9%), and torque teno midi virus (TTMDV) (22.8%) (Fig. 3b). To better elucidate which respiratory viruses are associated with the susceptibility to multiple ARTIs, we further used qPCR-based assays to delineate specific viruses in the virome by selecting the top three respiratory viruses and top three anelloviruses detected by NGS. Because early childhood infection by respiratory syncytial virus (RSV) or HRV had been reported to be associated with recurrent asthma[21,22], RSV was also subjected to RT-qPCR assay. Unexpectedly, neither the above respiratory viruses, nor

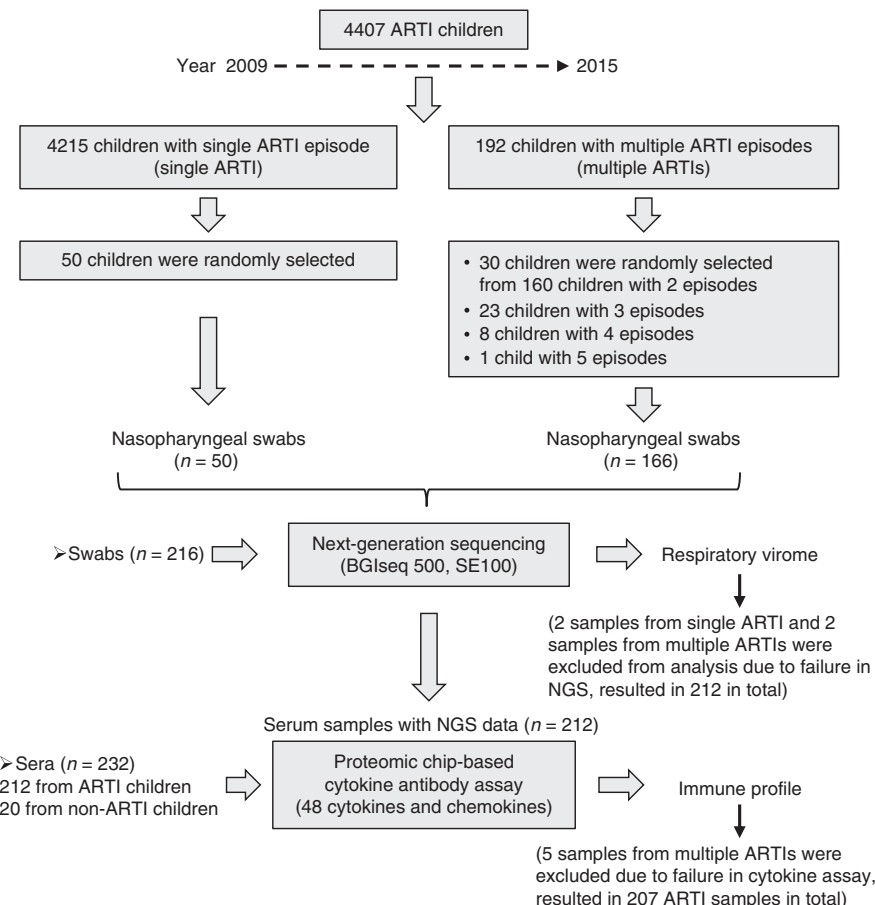

**Fig. 1** Flow chart of the study design. Children with multiple or single ARTIs were selected from 4407 ARTI children during 2009–2015 as described in the Materials and Methods. For respiratory virome analysis, a total of 164 and 48 swabs from children with multiple and single ARTIs were included, respectively. For the immune profile determination, corresponding samples including 159 (5 sera without sufficient amount were excluded from cytokine assay) and 48 sera from children with multiple and single ARTIs, respectively, together with 20 non-ARTI healthy children were finally included

those anelloviruses had increased detection in children with multiple ARTIs (Fig. 3c, d), suggesting no association of these viruses with multiple ARTIs.

**_Propionibacterium_ phages are associated with multiple ARTIs.** Further investigation of commensal bacteriophages revealed varied distribution of different phage taxa (according to their bacteria host tropism) in children with both single ARTI and multiple ARTIs (Fig. 4a). Decreased abundance of some phages, but increased abundance of others were observed in the multiple-ARTIs children (Fig. 4b). Among individual bacteriophage taxa, _Propionibacterium_ phages were the most abundant and having significantly higher levels in children with multiple ARTIs than those with a single ARTI for each additional episode of infection (Fig. 4c); other abundant phages such as the _Lactococcus_ phages, in contrast, showed a reverse trend (Fig. 4d and Supplementary Fig. 3). Moreover, the abundance of _Propionibacterium_ phages showed a progressively increasing trend along with the recurrence of ARTI episodes (Fig. 4c). In particular, about 64.0% of children with multiple ARTIs was positive for _Propionibacterium_ phages, significantly higher than the 6.3% in the single-ARTI children ($P < 0.001$, Fisher's exact test) (Fig. 4e). These results demonstrate a specific correlation between recurrent ARTIs and _Propionibacterium_ phages, and strongly suggest a role for _Propionibacterium_ phages as a potential factor for recurrent ARTIs.

**TIMP-1 and PDGF-BB are associated with multiple ARTIs.** Cytokines play an important role in mediating antiviral and anti-bacterial responses. To better understand the role of cytokines in ARTI, we measured 48 common cytokines and chemokines in sera obtained from the single-ARTI and multiple-ARTIs children using the Proteomic Chip-based Cytokine Antibody Assay. Only two cytokines, tissue inhibitor of metalloproteinases 1 (TIMP-1) ($P < 0.0001$) and platelet-derived growth factor subunits BB (PDGF-BB) ($P < 0.05$) showed significantly elevated levels in children with multiple ARTIs compared with children with either a single ARTI or no infection (Fig. 5a, b, and Supplementary Fig. 4, nonparametric Kruskal-Wallis Test). Moreover, both cytokines in children with multiple ARTIs remained higher than that in the single-ARTI and non-ARTI children for each episode of infection, suggesting the release of these cytokines being triggered constantly in vivo during recurrent ARTIs.

To examine whether these cytokine changes are cofounded by other possible risk factors for the occurrence of multiple ARTIs, cytokine levels, age, season of infection, and sampling time, were subjected to a multivariate logistic regression analysis with backward elimination. Because of an inevitable increase in the age of children at later episodes, we used both age at the first episode of infection and the median age of each child with multiple episodes over time into the analyses with two data normalizations: either Log2 or Z-score transformation. When the ages at the first episode of infection for the multiple-ARTIs

**Table 1 Comparison of demographic and clinical characteristics between children with single and multiple ARTIs**

| Parameters | Single ARTI[a] | Multiple ARTIs[a] | | | | P value[b] | P value[c] |
|---|---|---|---|---|---|---|---|
| | | 1st ARTI | 2nd ARTI | 3rd ARTI | 4th ARTI | | |
| Demographics | (n = 48) | (n = 61) | (n = 61) | (n = 32) | (n = 10) | NA | NA |
| Median age (month, range) | 40.8 (4.8–63.6) | 45.6 (9.6–114) | 52 (12–122) | 72 (28–142) | 60.5 (41–97) | 0.082 | 0.000 |
| Gender (M/F) | 23/25 | 27/34 | 27/34 | 15/17 | 3/7 | 0.847 | 0.855 |
| Duration of sampling[&] | 2.3 (0–13) | 2.1 (0–14) | 2.2 (0–14) | 2.3 (0–15) | 2 (0–4) | 0.426 | 0.631 |
| **Season** | (n = 48) | (n = 61) | (n = 61) | (n = 32) | (n = 10) | NA | NA |
| Spring | 5 (10.4%) | 13 (21.3%) | 18 (29.5%) | 16 (50.0%) | 2 (20.0%) | 0.192 | 0.037 |
| Summer | 4 (8.3%) | 12 (19.7%) | 11 (19.0%) | 4 (12.5%) | 2 (20.0%) | 0.108 | 0.830 |
| Autumn | 24 (50.0%) | 20 (36.8%) | 18 (32.8%) | 5 (15.6%) | 5 (50.0%) | 0.112 | 0.141 |
| Winter | 15 (31.3%) | 16 (26.2%) | 14 (26.2%) | 7 (21.9%) | 1 (10.0%) | 0.669 | 0.801 |
| **Clinical parameters** | (n = 48) | (n = 61) | (n = 61) | (n = 32) | (n = 10) | NA | NA |
| Cough | 36 (72.0%) | 53 (86.9%) | 53 (86.9%) | 21 (65.6%) | 7 (70.0%) | 0.138 | 0.035 |
| Fever | 44 (88.0%) | 58 (95.1%) | 55 (90.2%) | 30 (93.8%) | 9 (90.0%) | 0.697 | 0.665 |
| Sore throat | 14 (28%) | 28 (45.9%) | 19 (31.2%) | 17 (53.1%) | 2 (20.0%) | 0.112 | 0.081 |
| Running nose | 29 (58.0%) | 49 (80.4%) | 40 (65.6%) | 19 (59.4%) | 9 (90.0%) | 0.032 | 0.064 |
| Expectoration | 18 (36.0%) | 23 (37.7%) | 13 (21.3%) | 7 (21.9%) | 4 (40.0%) | 1.000 | 0.144 |
| White blood cell ($10^9$/L) | 9.22 (4.55–22.69) | 8.65 (3.96–20.3) | 7.77 (2.85–39.1) | 8.43 (3.64–20.1) | 6.97 (4.0–14.3) | 0.557 | 0.455 |
| Lymphocyte (%) | 26.1 (0.29–74.7) | 29.7 (1.9–86.2) | 28.4 (2.1–71) | 23.5 (5.0–69.5) | 24.2 (15.9–41.8) | 0.688 | 0.609 |
| Neutrophils (%) | 62.5 (0.55–84.1) | 59.7 (12.2–86.3) | 58.2 (2.8–86.5) | 67 (19.5–92.5) | 64 (51.4–80.2) | 0.920 | 0.097 |
| Hemoglobin (g/L) | 125.5 (96–230) | 123 (95–136) | 123 (103–149) | 120 (94–143) | 122 (114–138) | 0.143 | 0.132 |
| Platelet ($10^9$/L) | 250 (139–745) | 246 (94–421) | 252 (150–398) | 256 (172–416) | 254 (157–343) | 0.467 | 0.917 |
| **Diagnosis** | n = 17 (48) | n = 25 (61) | n = 30 (61) | n = 29 (32) | n = 8 (10) | NA | NA |
| Tonsillitis | 6 (35.3%) | 13 (52.0%) | 17 (56.6%) | 20 (68.9%) | 5 (62.5%) | 0.353 | 0.629 |
| Pharyngitis | 1 (5.8%) | 2 (8.0%) | 0 | 1 (3.4%) | 0 | NA | NA |
| Bronchiolitis | 8 (47.1%) | 8 (32.0%) | 14 (46.6%) | 8 (27.6%) | 4 (50.0%) | 0.353 | 0.373 |
| Pneumonia | 2 (11.8%) | 3 (12.0%) | 0 | 1 (3.4%) | 0 | NA | NA |

For other details, please see Fig. 1 and Methods
M male, F female, NA not applicable
[a]One child having two ARTI episodes and two children having only one ARTI episode were excluded from the final analysis due to failure in NGS, and thus results from a total of 61 multiple-ARTI children and 48 single-ARTI children were presented
[b]Fisher's exact test and Mann–Whitney U test were used for the comparison of single ARTI and the first episode in multiple ARTIs
[c]Fisher's exact test and Kruskal Wallis test were used for the comparison of different episodes in multiple ARTIs. & Days between disease onset and sampling

**Table 2 Risk factors associated with multiple ARTIs**

| Predictive variables[a] | P value | Adjusted OR | 95% CI | |
|---|---|---|---|---|
| | | | Lower | Upper |
| Log2 TIMP-1 | <0.001 | 61.56 | 8.19 | 462.87 |
| Log2 PDGF-BB | 0.005 | 11.58 | 2.08 | 64.61 |
| Age | 0.039 | 1.51 | 1.02 | 2.24 |
| Log2 ICAM-1 | 0.078 | 1.57 | 0.95 | 2.61 |

The results of other three similar analyses were shown in Supplementary Tables 2–4
OR odds ratio, CI confidence interval
[a]Serum levels of the cytokines were log2-transformed and then analyzed using logistic regression model with backward elimination

group and Log2 transformation were used, two cytokines TIMP-1 ($P < 0.001$) and PDGF-BB ($P = 0.005$), as well as age ($P = 0.039$) were identified with significant predictive values (Table 2). Moreover, TIMP-1 and PDGF-BB had adjusted odds ratios (OR) of 61.56 and 11.58, respectively; whereas age did not substantially increase the OR values (Table 2).

Similar results were observed in the analysis with Z-score transformation (Supplementary Table 2). In particular, when the median age of subjects was used, only TIMP-1 ($P < 0.001$) and PDGF-BB ($P = 0.005$) were identified with significant predictive values (Supplementary Table 3, 4). All above logistic regression analyses verified that the two cytokines TIMP-1 ($P \leq 0.001$; adjusted ORs: 5.05–61.56) and PDGF-BB ($P$ values: 0.078–0.003; adjusted ORs: 2–11.58) are significantly associated with the occurrence of multiple ARTIs. Age, despite also being a potential cofounding factor, has relatively weaker association with multiple ARTIs. These data indicate that the serum levels of these two cytokines may have diagnostic values for the occurrence of multiple ARTIs.

**Association between TIMP-1, PDGF-BB and bacteriophages.** Having demonstrated that both *Propionibacterium* phages and cytokines TIMP-1 and PDGF-BB are associated with multiple ARTIs, we wondered whether the productions of TIMP-1 and PDGF-BB and the presence of *Propionibacterium* phages are linked, in order to generate a unified hypothesis to explain their respective roles in the pathogenesis of multiple ARTIs. Significantly higher levels of both cytokines were observed in *Propionibacterium* phages-positive children than *Propionibacterium* phages-negative children ($P < 0.01$) (Fig. 5c, d, Mann–Whitney U test); whereas other phages such as the non-pathogenic *Lactococcus* had no association with the production of either cytokines (Fig. 5e, f). Because neither the presence of common respiratory viruses, nor anelloviruses, showed significant correlation with the elevated productions of these two cytokines (Supplementary Fig. 5), it is reasonable to speculate that the productions of TIMP-1 and PDGF-BB were triggered by non-viral factors. Consistently, the presence or absence of any other type of viruses did not significantly affect the

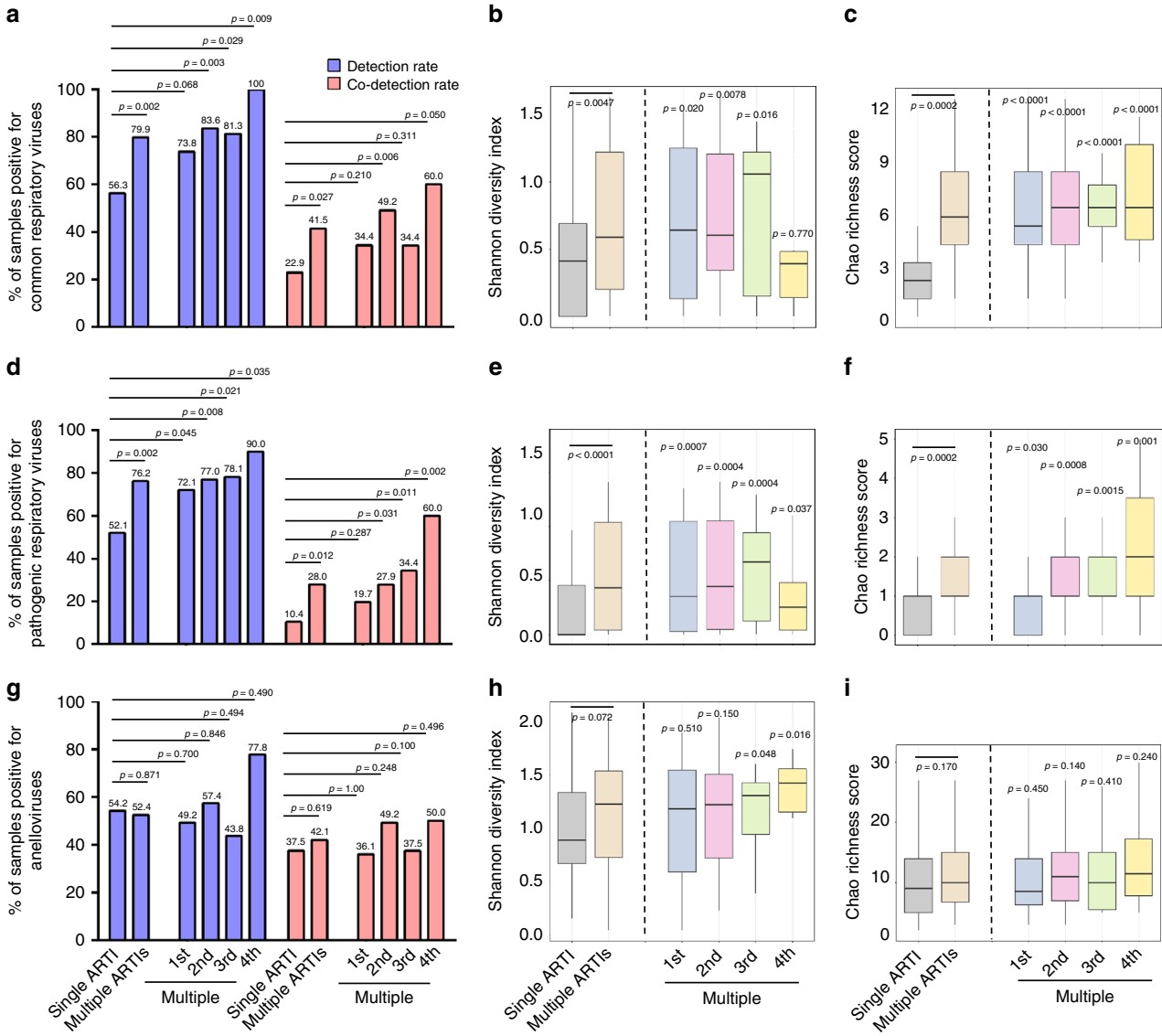

**Fig. 2** Characteristics of respiratory virome among children with single or multiple ARTIs. Detection and co-detection rates were compared by Fisher's exact test for common respiratory viruses (**a**), pathogenic respiratory viruses (excluding the persistent DNA viruses: herpesviruses, polyomaviruses, papillomaviruses, species C adenoviruses and adeno-associated virus) (**d**), and anelloviruses (**g**). The viral alpha diversity (Shannon Diversity Index) were analyzed using nonparametric Kruskal-Wallis Test correcting for multiple comparisons with Dunn's procedure for common respiratory viruses (**b**), pathogenic respiratory viruses (**e**) and anelloviruses (**h**). The viral richness (Chao Richness score) were analyzed using nonparametric Kruskal-Wallis Test corrected for multiple comparisons with Dunn's procedure for common respiratory viruses (**c**), pathogenic respiratory viruses (**f**), and anelloviruses (**i**). The number of samples in each group were 48 (single ARTI), 164 (multiple ARTIs), 61 (1st), 61 (2nd), 32 (3rd), 10 (4th) (as only one sample was in the 5th episode, we combined the 5th episode into the 4th episode group). The results were presented as box and whiskers plots. The central line shows the median, and whiskers above and below the box show the minimum and maximum within 1.5 interquartile range (IQR) of the lower quartile and the upper quartile

association between the presence of *Propionibacterium* phages and the increased serum levels of TIMP-1 and PDGF-BB (Supplementary Fig. 6).

To evaluate the diagnostic potential of the three factors on the recurrence of ARTIs, we performed the standard receiver operating characteristic (ROC) curve analyses. Results showed that TIMP-1 and PDGF-BB had area under the curve (AUC) of 0.78 and 0.72, respectively (Fig. 6a). The combination of both cytokines improved AUC to 0.81, and these two cytokines plus *Propionibacterium* phages further improved AUC to 0.94 (Fig. 6a), demonstrating these three markers have excellent discriminative power between multiple ARTIs and single ARTI. Similar results were obtained using non-ARTI group as a control for the multiple-ARTIs group (Fig. 6b), again demonstrating that

together these three factors have diagnostic potential for multiple ARTIs.

## Discussion

ARTIs are thought to be mainly caused by some common respiratory viral and bacterial pathogens[1,2]; however, specific mechanisms responsible for frequent recurrent ARTIs are ill defined[3,4]. Routine treatment for ARTIs with broad-spectrum antibiotics in China, and perhaps elsewhere[23], often lacks diagnostic evidence. This practice results in antibiotics misuse or overuse, which drives the generation and spreading of drug resistance[24]. Discovering specific pathogens causing recurrent ARTIs is therefore important for targeted therapy. To this end, we

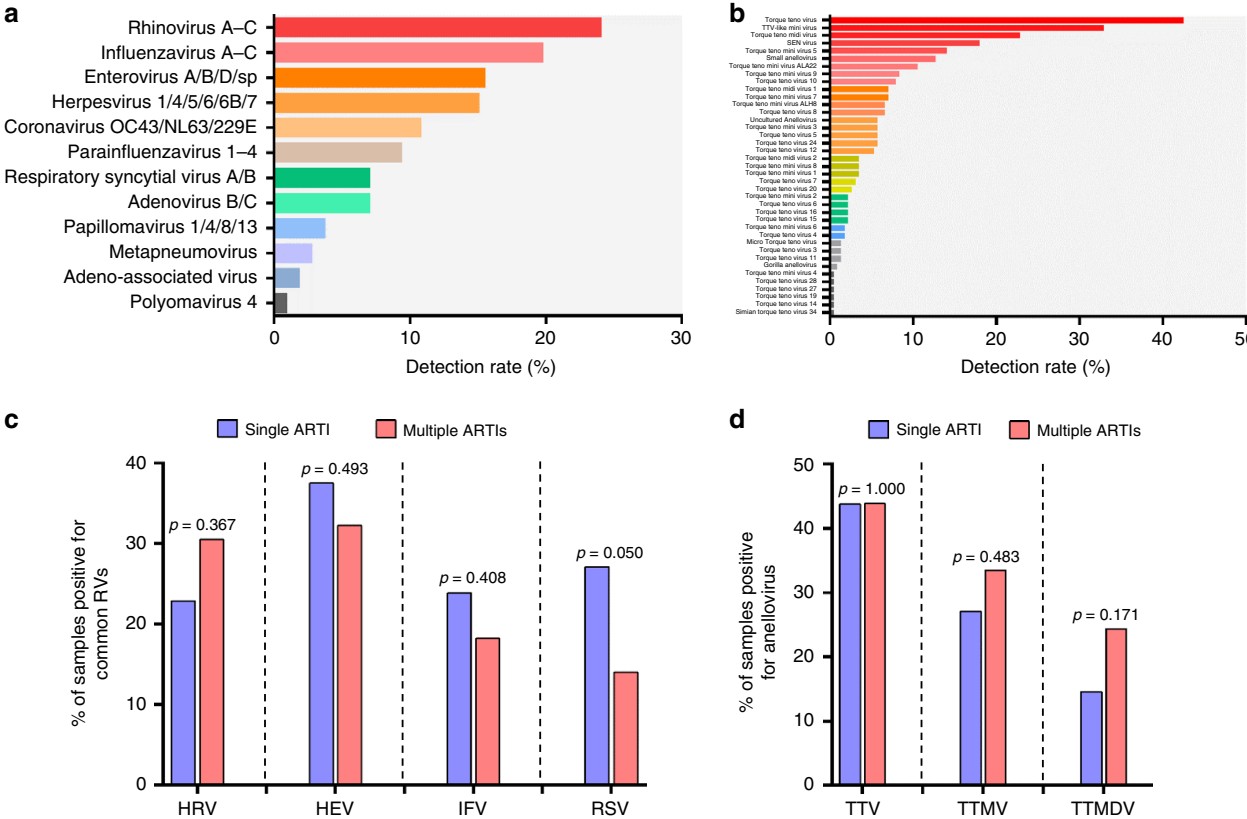

**Fig. 3** Detection rates of eukaryotic viruses among children with ARTIs. Detection rates based on metagenomics for respiratory viruses (**a**), and anelloviruses (**b**), among children with ARTIs. Comparison of detection rates as determined by PCR-based assays between single-ARTI (blue) and multiple-ARTIs children (red) for four most common pathogenic respiratory viruses (i.e., HRV, HEV, IFV and RSV) (**c**), and three main anelloviruses (i.e., TTV, TTMV and TTMDV) (**d**). In addition to the top three most abundant viruses (HRV, HEV, IFV), RSV was also included as a control because it is one of the major causes of lower respiratory tract infections during early life, and it is associated with recurrent asthma in later life

have made a number of findings that are of potential medical significance using a combination of NGS, cytokine profiling, and multivariate analyses.

Changes in virome characteristics have been reported to be associated with various diseases, such as inflammatory bowel disease, AIDS, enteric graft-versus-host disease, cancer, type I diabetes, etc.[25–30]. As a main component of the respiratory virome, bacteriophages can affect human health by either shaping respiratory microbiota or directly interacting with the immune system[8,31,32]. Their frequency and diversity may vary to adapt external and internal environments[33], but relatively few studies have shed lights onto bacteriophages when parsing respiratory virome (Supplementary Table 5)[7,8,34]. One of the major findings of this study is the association between *Propionibacterium* phages and multiple ARTIs in children (Fig. 4). To our knowledge, this study is the first to demonstrate an association between *Propionibacterium* phages and multiple ARTIs.

Specialized bacterial communities in the respiratory tract are thought to be the gatekeepers for respiratory health[34]. Respiratory microbiota can maintain the structural maturation and integrity of the respiratory tract by preventing pathogen infection and shaping local immune responses. Bacteriophages shape the composition, diversity and function of bacteria community by various ways, and their types are strong correlated with specific bacteria species[35–37]. The increased presence of *Propionibacterium* phages should reflect the existence of more *Propionibacterium*, and the decreased abundance of *Lactococcus* phages and other phages may imply a decreased density of *Lactococcus* and other bacteria.

*Propionibacterium* is a commensal bacteria in the microbial community of the skin, oral/nasal cavity, and gastrointestinal and genitourinary tracts[38]. Some *Propionibacterium* species such as *Propionibacterium acnes* can cause skin damage (e.g., acne lesions), and they have been associated with diseases such as sciatica and implant-associated infections[38,39]. In respiratory tract, *Propionibacteria* had been linked to ventilator-associated pneumonia, chronic rhinosinusitis, virus infection[40–43], as well as pulmonary inflammation[44]. *Lactococcus* are generally regarded as a type of benign commensal bacteria in humans[45], and often reported to decrease in patients with an imbalanced microbiota[46,47]. High frequency of *Propionibacterium* phages, together with low frequencies of *Lactococcus* phages and most other bacteriophages, in upper respiratory tract of children with multiple ARTIs over time is indicative of a respiratory microbiota imbalance between beneficial and pathogenic bacteria (Fig. 4b–e, Fig. S3), and such an imbalance may render airway to be more susceptible to infection by respiratory viruses or colonization by pathogenic bacteria (Figs. 2a–c and 4c). Colonization of *Propionibacteria* or other pathogenic bacteria could enhance the airway inflammation[48,49], which could make individuals prone to recurrent airway infections. Consistent with this idea, children with multiple ARTIs were found to have higher rates of detection and co-detection of pathogenic respiratory viruses with higher diversity and richness than children with single ARTI (Fig. 2d–f), whereas the same trend was neither observed for non-pathogenic *Anelloviridae* family (Fig. 2g–i), nor for three definitive bacterial pathogens (*Streptococcus pneumoniae*, *Haemophilus influenzae*, and *Staphylococcus aureus*) (Supplementary Table 6). These

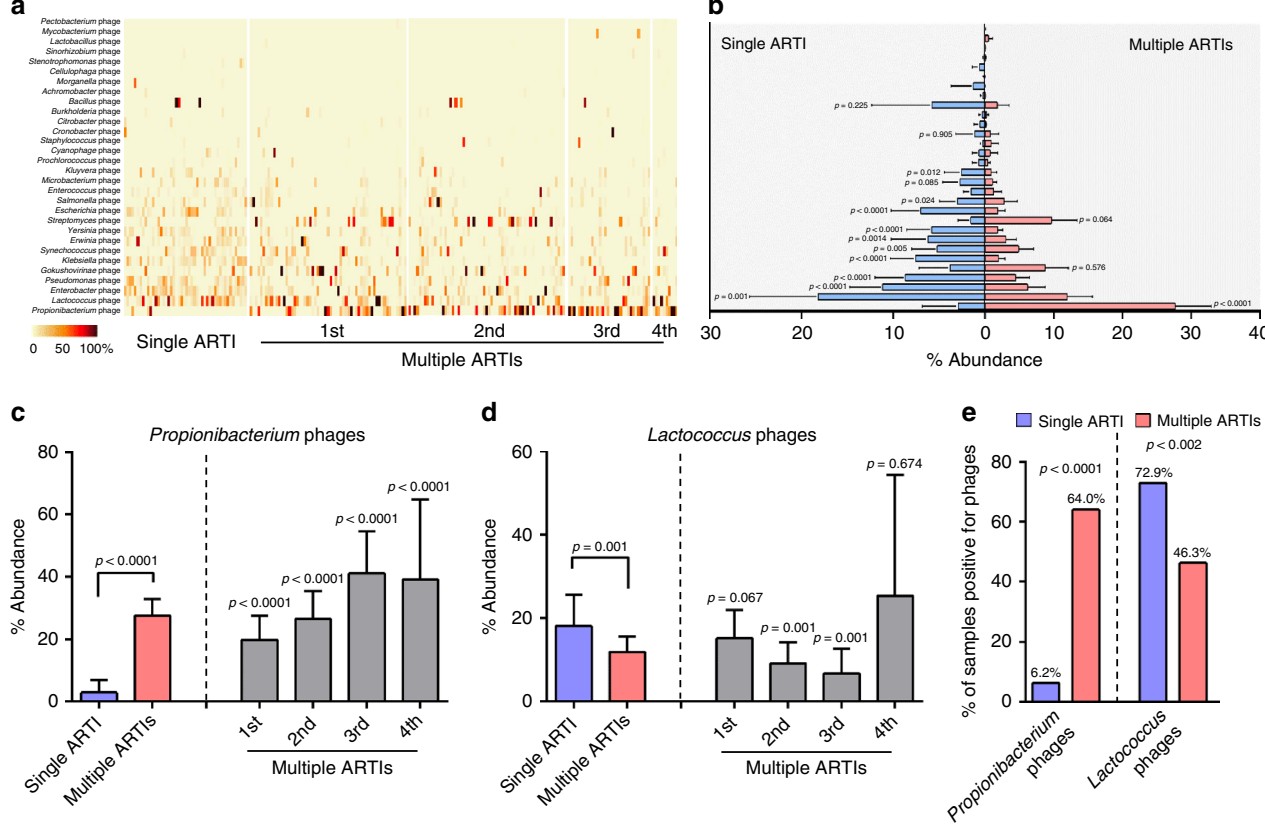

**Fig. 4** Comparison of bacteriophage profile between single-ARTI and multiple-ARTIs. **a** Bacteriophage community profile was displayed as a heatmap of the relative abundance, which included the 30 most frequently detected bacteriophage taxa, classified according to their bacteria host and grouped by ARTI episodes. Relative abundance was calculated based on the proportion of normalized reads number. The color gradient key displays percent abundance. **b** Comparison of the relative abundance of each bacteriophage taxa between children with single and multiple ARTIs using Mann–Whitney U test. Each column displays the mean abundance with the 95% CI. Comparison of abundance of *Propionibacterium* phages (**c**), and *Lactococcus* phages (**d**), between multiple-ARTIs children and single-ARTI children were performed using nonparametric Kruskal-Wallis Test corrected for multiple comparisons with Dunn's procedure. Each column displays the mean abundance with the 95% CI. **e** Comparison of positive rates of *Propionibacterium* phages and *Lactococcus* phages between children with single and multiple ARTIs was performed using the Fisher's exact test. The number of patients in each group in panels of **a**, **c**, and **d** was the same with those in Fig. 1

suggest that the microbial dysbiosis in the upper respiratory track leads to greater susceptibility to infection and hence the occurrence of multiple ARTIs children.

Viral-bacterial symbiosis in the respiratory tract is one of the main mechanisms causing respiratory diseases, in which respiratory viruses promote bacterial colonization, or reversely, respiratory bacteria promote viral infection through various pathways[8]. Intriguingly, except for *Propionibacterium* phages, known pathogenic respiratory viruses (Fig. 3c), non-pathogenic *anelloviruses* (Fig. 3d), as well as bacterial pathogens such as *S. pneumoniae*, *H. influenza*, and *S. aureus*, did not show a significant correlation with multiple ARTIs. Therefore, high airway susceptibility of the multiple-ARTIs children to pathogen infection might have resulted from the combined effects of an unstable respiratory microbial communities and excessive airway inflammation induced by bacterial pathogens or bacteriophages themselves[50–52].

Another interesting finding of this study is that elevated serum levels of TIMP-1 and PDGF-BB are significantly associated with multiple ARTIs (Fig. 5a, b). TIMP-1 is a member of the natural inhibitors of matrix metalloproteinases (MMPs), and it is produced by fibroblasts, epithelial cells, endothelial cells, and smooth muscle cells[53]. PDGF-BB is a growth factor mainly secreted by eosinophils and platelets[54]. Both TIMP-1 and PDGF-BB have

been implicated in asthma and chronic obstructive pulmonary disease (COPD)[55,56], in which there was characteristic destruction of the extracellular matrix (ECM) and tissue remodeling of sub-epithelial mesenchymal cells. Mechanistically, elevated levels of TIMP-1 and MMP-9 can induce profibrotic environment and airway remodeling by causing aberrant ECM degradation or accumulation of ECM proteins in pulmonary alveoli and small airway walls, leading to COPD[57]. PDGF-BB is a major smooth muscle cells (ASM) mitogen, the overexpression of which can induce airway hyper-responsiveness and remodeling, and decrease lung compliance by stimulating the ASM proliferation and increasing airway fibroblast activity[54,58,59]. Although increased level of TIMP-1 or PDGF-BB was observed among COPD and asthma patients[55,56,59–61], only a few children in our cohort had documented asthma (2 cases in the single-ARTI group, 3 cases in the multiple-ARTIs group), and no child had COPD. The low percentage (~2.3%) of asthma in our study cohort suggested that asthma or COPD had little influence on the occurrence of multiple ARTIs in children. Instead, our finding of increased level of TIMP-1 or PDGF-BB in children with multiple ARTIs over time strongly suggested a function that both TIMP-1 and PDGF-BB play in recurrent respiratory infections, and thus highlighted the urgency for studying these two cytokines in order to discover potential therapeutic strategies[55,56].

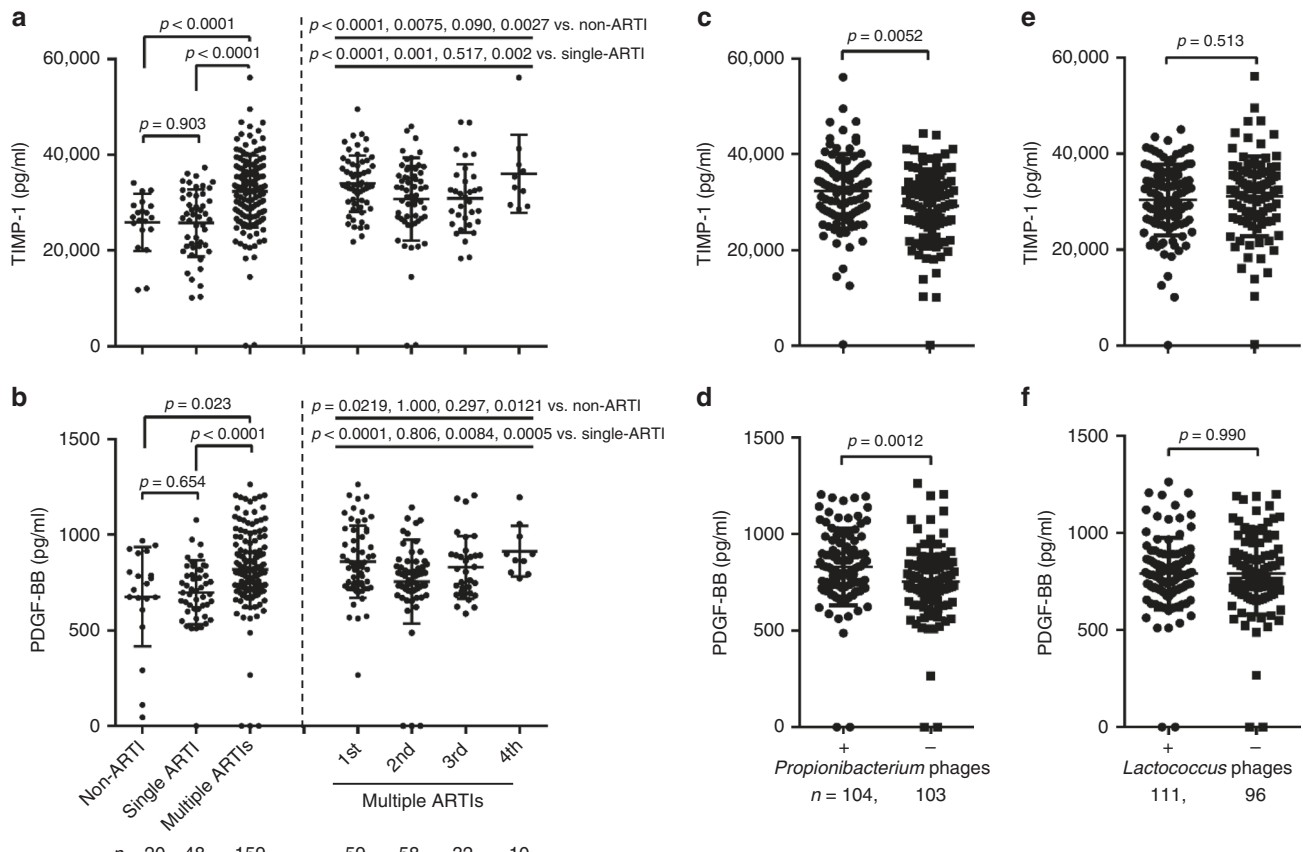

**Fig. 5** Association between cytokine levels and bacterial phage in different groups. **a**, **b** The serum levels of TIMP-1 and PDGF-BB between children with single and multiple ARTIs were analyzed using nonparametric Kruskal-Wallis Test corrected for multiple comparisons with Dunn's procedure. **c**, **d** Comparison of TIMP-1 and PDGF-BB levels between children who were positive and negative for *Propionibacterium* phages was made by using Mann–Whitney *U* test. (**e**, **f**) Comparison of TIMP-1 and PDGF-BB levels between children who were positive and negative for *Lactococcus* phages was made by using Mann–Whitney *U* test. Error bars indicate the standard deviation. Results of all other cytokines are shown in supplemental data (Supplementary Fig. 4). Because five sera from the multiple-ARTIs groups were insufficient for cytokine analyses, there were five fewer patients in this group than that in Figs. 1 and 3

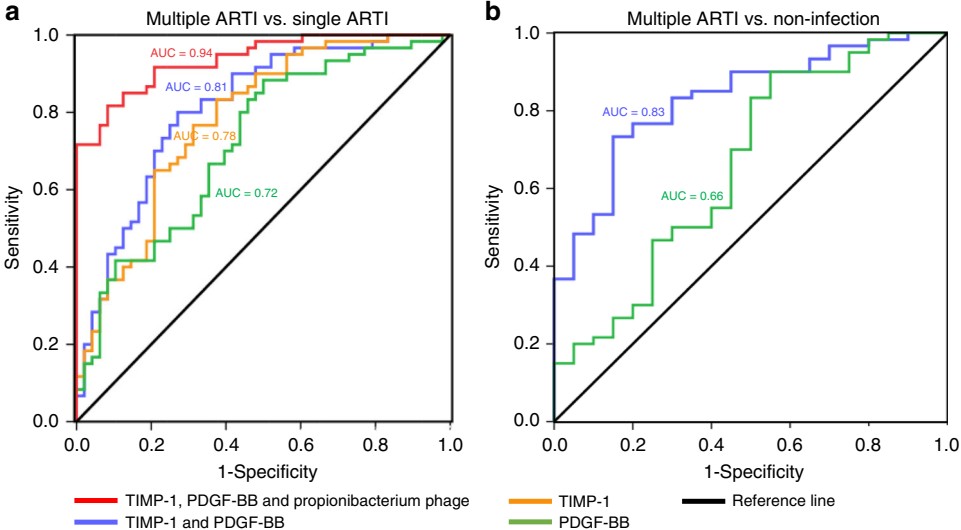

**Fig. 6** TIMP-1, PDGF-BB, and *Propionibacterium* phages are predictor for multiple ARTIs. **a** ROC curves with data comparing multiple ARTIs and single ARTI. **b** ROC curves with data comparing multiple ARTIs and non-ARTI healthy controls. Because there was no phage data for non-ARTI healthy controls, only TIMP-1 and PDGF-BB were used in the comparison between multiple ARTI children and non-ARTI children. Because the curves for both TIMP-1 and PDGF-BB, and for TIMP-1 alone were fully overlapped, only the former was shown (blue curve)

It is unclear what stimulated the expression of TIMP-1 and PDGF-BB in the children with multiple ARTIs. Intriguingly, the presence of *Propionibacterium* phages, rather than common pathogenic respiratory viruses or non-pathogenic *anelloviruses*, was closely associated with higher levels of TIMP-1 ($P < 0.01$) and PDGF-BB ($P < 0.001$) (Fig. 5c–f, Figs. S5 and S6, Mann–Whitney U test). *Propionibacteria* such as *propionibacterium acnes* was demonstrated to induce high proinflammatory cytokine levels and lead to chronic skin disease[62,63]. The same cytokine-inducing mechanism operating in *P. acnes* might also be in operation in ARTIs. Based on our data and other published information, we suggest a mechanism in which *Propionibacteria* up-regulates the expressions of both TIMP-1 and PDGF-BB; TIMP-1 prevents local degradation of ECM proteins and tissue remodeling, and PDGF-BB promotes ACM proliferation and airway hyper-responsiveness; together, these factors cause airway remodeling and epithelium inflammation, and thus render easier cell penetration by pathogenic respiratory viruses, and further exacerbate dysbiosis (Supplementary Fig. 7).

Other than providing some mechanistic insight into the possible interplay between host and pathogens during recurrent respiratory infections, our findings of the association between the levels of TIMP-1, PDGF-BB, *propionibacterium* phages and the occurrence of multiple ARTIs could have an important use for early diagnosis, which has public health potential. Both logistic regression and ROC curve analyses suggested that the serum levels of TIMP-1 (OR: 61.56 and AUC: 0.78) and PDGF-BB (OR: 11.58 and AUC: 0.72) are strong indicators for predicting the occurrence of multiple ARTIs (Table 2 and Fig. 6). When combined with *Propionibacterium* phages, the three factors could obtain the highest prediction of sensitivity and specificity for susceptibility to airway infections (AUC: 0.94) (Fig. 6). If early diagnosis could be accomplished, immediate separation of a susceptible child from an infectious environment may help to reduce the risk of the child from suffering recurrent airway infections.

We acknowledge that this study has several limitations. First, this is a retrospective study. Even though we employed a stringent selection criterion, we were unable to fully exclude the possibility that a few children in the single-ARTI group might have experienced multiple ARTIs, but were not diagnosed as such because patients went to different hospitals, took self-medication at home, or episodes occurred before the enrollment into this study or after the completion of this study. In spite of all these perceived limitations, the multiple-ARTIs children were characterized by significantly elevated serum TIMP-1 and PDGF-BB levels and increased abundance of respiratory *Propionibacterium* phage in comparison to the single-ARTI and other control groups, implying the number of cases misclassified, if any, was small. A well-designed prospective study will help to further confirm these findings reported here, and to fully assess the predictive power of TIMP-1 and PDGF-BB, as well as *Propionibacterium* phage for susceptibility to multiple ARTIs. Second, the medical history of asthma or allergic disease of the selected children was not available, and eosinophils and IgE levels were not measured in this study. Some previous studies have demonstrated an association between TIMP-1 or PDGF-BB and asthma in children[60,61]. Whether there is an association between frequent ARTIs and asthma needs to be answered in the future. Third, multiple displacement amplification (MDA) with phi29 DNA polymerase was widely used to generate DNA templates for NGS[25,28,64]. However, phi29 DNA polymerase can lead to preferential amplification of circular DNA templates[65–67]. Therefore, the use of phi29 DNA polymerase might lead to a small bias toward some *Propionibacterium* phage with circular genomes. Furthermore, because this study was not designed for the discovery of highly divergent and novel human viruses, we used nucleic acid-based method which has high specificity, rather than translated alignments that allows for broad coverage, for the virus annotation. Our high specificity strategy may be unable to define unclassified respiratory viruses. Finally, we failed to obtain data on the respiratory microbiota due to small sample amounts and other technical reasons, which limited our ability to directly compare the composition of the respiratory microbiota between children with multiple ARTIs and single ARTI, and restricted our capacity to provide direct evidence of an imbalanced microbiota over time among the children with multiple ARTIs.

In summary, we demonstrated in 109 clinically defined ARTI children carefully selected from a large pediatric community cohort of more than four thousands subjects and 20 healthy controls that serum levels of TIMP-1 and PDGF-BB as well as tissue abundance of *Propionibacterium* phages are significantly associated with recurrent ARTIs. The combination of these three factors might have important diagnostic potential for predicting frequent ARTIs in children, and such prediction may be used as a guidance for initiating early intervention.

## Methods

**Ethics statement**. This study was approved by the Ethics Committees of Shanghai Public Health Clinical Center and Shanghai Nanxiang Hospital, and complied with all relevant ethical regulations. Oral or written informed consents were obtained from children's parents or guardians before sample collection.

**Study design and human subjects**. This retrospective study included 4,407 children diagnosed as having ARTI by clinicians during a period of six years from 2009 to 2015 at Nanxiang Hospital in Shanghai, China. All children were from the Jiading District of Shanghai, and shared the same weather and environmental condition. The diagnostic criteria of ARTI include (1) having a body temperature above 37.5 °C; and (2) having at least one of the following symptoms and signs: cough, sore throat, running nose, and expectoration. Alternatively, having a clinical diagnosis of tonsillitis, pharyngitis, bronchiolitis and pneumonia. Demographic and clinical features were recorded by the specialists in the hospital (Table 1). In children in whom informed consents were obtained from their parents, nasopharyngeal swabs were collected and then stored in 3 ml virus transport medium (Yocon, Beijing, China), and serum samples were collected from venous blood. The sample collection was performed by clinical specialists at first visit to the hospital, and children did not receive any medical treatment including antibiotics treatment at the time of sample collection. Furthermore, as a prescription drug, antibiotics are not available to the general public without medical record, especially in Shanghai where the rules and regulations are strictly followed. Therefore, the possibility of self-medication for the children with ARTIs before they visited the hospital was small. All the samples were transported to laboratories at Institut Pasteur of Shanghai, stored at −80 °C until the experiments were performed.

**Selection criteria of single and multiple ARTIs**. To identify children who experienced two or more episodes of ARTIs (herein defined as multiple ARTIs) during the study period between 2009 and 2015, we first reviewed patients' medical records for having identical name and birth date; or having identical parents' name and home address; or having identical contact telephone number and home address. Becasue the majority of the respiratory viral infections only remain PCR positive for less than 2 weeks, we defined the multiple ARTI cases as having at least two consecutive ARTI episodes that were separated by an interval of at least 30 days. Patients who did not meet the definition of multiple ARTIs were classified as having single ARTI.

Using the above criteria, we identified 192 multiple-ARTIs children. Among them, 160 children had two ARTI episodes, 23 had three, 8 had four and one child had five episodes. To investigate the respiratory virome associated with multiple ARTIs, we randomly selected 30 children experienced two episodes and all 32 other children experienced three to five episodes for metagenomic sequencing (Fig. 1). The median time intervals for secondary, third and fourth ARTI episode were 7.2 (range, 1.0–38.4), 15.6 (1.0–44.4), and 8.4 (1.1–18) months, respectively. For comparison, 50 children who experienced only one ARTI episode during the study period (known as the single-ARTI group) were randomly selected from each year. The single-ARTI group matched the multiple-ARTIs group with respect to age, gender, clinical presentations, laboratory test results, as well as the seasonal distribution (Table 1).

**Processing of swab sample for metagenomic sequencing**. An established viral metagenomics method for enrichment of encapsidated DNA and RNA viruses was used for sample pretreatment[27,68]. Briefly, 200 μl homogenized suspension was

centrifuged for 15 min at $12,000 \times g$ at room temperature. Supernatant was filtered through a 0.45 μm sterile filter (Costar Spin-X centrifuge tube filters, Corning, USA), and cell-free nucleic acids in the filtrate was digested by incubation with a cocktail of nucleases including 15U Turbo DNase (Invitrogen, USA), 20U Benzonase (Novagen, Germany) and 20U RNase I (Promega, USA) for 2 h, at 37 °C.

**Nucleic acid extraction and pre-amplification**. Total nucleic acids (including DNA and RNA) were extracted using QIAamp MinElute virus kit (Qiagen, Germany), and then amplified using a random-amplification approach (REPLI-g Single Cell WTA kit, Qiagen, Germany) for metagenomic analysis[69,70]. Briefly, the method uses the MDA, an efficient method to amplify small amounts of DNA samples. According to the manufactures' instruction provided with the WTA kit, nucleic acids were first reverse-transcribed to complementary DNA (cDNA) using random primers, and then subjected to a ligation reaction to obtain large-fragments of DNA. The obtained DNA was further amplified using MDA with phi29 polymerase for 2 h. The amplified products were used for library construction and sequencing.

**Library construction and next generation sequencing (NGS)**. The genomic library construction and NGS were conducted by BGI using the BGI-Seq500 sequencing platform (National Gene Bank, BGI, China)[71]. The MDA products were first purified using AmPure beads (Qiagen, Germany), and quantitated by Qubit Fluorometer 3.0 (Life Technologies, USA) and Agilent 2100 Bioanalyzer (Agilent Technologies Inc. USA). A total of 1 μg purified products were fragmented to 250 bp by Covaris E210 (Covaris, USA), followed with an end repairing by adding 3′-adenine overhangs. The short DNA fragments were ligated to BGI-Seq500 Ad153 2B adapter, followed by 12 cycles of amplification. Amplified products were further purified by AmPure beads (Qiagen, Germany). After quantification by Qubit Fluorometer 3.0 (Life Technologies, USA), the products were pooled to make a single strand DNA circle (ssDNA circle). The sequencing substrate DNA nanoballs (DNBs) were generated with the ssDNA circle by rolling circle amplification[72]. The DNBs were loaded on the patterned nanoarrays and sequenced on BGI-Seq500 platform (BGI, China) using a single-end 100 bp sequencing.

One child having two ARTI episodes and two children having only one ARTI episode were excluded from the final analysis due to failure in NGS, and thus results from a total of 61 multiple-ARTI and 48 single-ARTI children were presented (Table 1).

**Bioinformatics analysis**. NGS raw data produced by the BGI-Seq500 platform was analyzed using a modified in-house pipeline based on the sequence-based ultra-rapid pathogen identification (SUPPI) principle[73]. The pipeline includes five steps. Step 1: quality control of reads. The raw data were filtered by SOAPnuke software to remove reads with sequencing adaptors, as well as low-quality (reads with more than 50% Phred quality score < 5 bases, or with more than 10% ambiguous base) and low-complexity reads[74]. Step 2: removing host sequences. Reads belonging to human host were subtracted from the data sets by mapping the reads to comprehensive human sequence databases using SNAP software[75]. The following three main human sequence databases (including one of Chinese ethnicity) were used:

HG19 (http://hgdownload.soe.ucsc.edu/goldenPath/hg19/bigZips/hg19.2bit),
YH (http://yh.genomics.org.cn/download.jsp),
refMrna (http://hgdownload.soe.ucsc.edu/goldenPath/hg19/bigZips/refMrna.fa.gz). Step 3: classification. The remaining reads were taxonomically classified to microbes using KRAKEN software by mapping to a modified reference database which contains viruses, archaea, fungi and plasmid (Supplementary Table 7). Step 4: verification. The above identified reads were aligned to the NCBI non-redundant nucleic acids database (May 2016) by BLAST (E-value < 0.001). The reads that map not only to viruses but also other species were discarded. In order to remove the false positive viral reads and achieve more accurate identification, all viral reads of each family were assembled into contigs by Minimo[76] and IDBA-UD[77]. The contigs and unassembled reads were blasted against the NCBI NT database. All identified viral sequences were kept and classified to the appropriate taxonomy levels (family, genus and species). Step 5: filtering. To reduce the potential false positives caused by the bleed over from other samples on the flow cell or environment[20,78,79], positive sample was defined only if the number of reads matching to virus references exceeded 15, which was based on a recent study[27]. The reads were normalized by raw data number of each sample (number of the preprocessed reads).

**Nucleic acid extraction and detection (HRV, HEV, IFV, RSV)**. Because the cutoff of 15 reads was rigorous enough to remove false positive results, but possibly result in false negative results, we selected the top three respiratory viruses and the top three anelloviruses identified among the children with ARTIs by NGS for further verification using qPCR-based assays. RNA was extracted from 200 μl nasopharyngeal swabs using Trizol LS reagent (Invitrogen, USA), and then reverse transcribed to cDNA using SuperScript III First-Strand Synthesis kit (Invitrogen, USA). The detection of HRV was conducted by a nested PCR method, the PCR conditions for both reactions were as follows: 30 cycles at 94 °C for 30 s, 50 °C for 30 s, and 72 °C for 90 s, followed by a final extension at 72 °C for 5 min. HEV, IFV and RSV

were detected using an in-house qPCR assay with specific primers (see Supplementary Table 8). The PCR conditions for qPCR were as follows: 94 °C 2 min, 45 Cycles at 94 °C for 15 s, 60 °C for 15 s, 72 °C for 30 s.

**Proteomic chip-based cytokine assay**. A total of 212 ARTI serum samples with available NGS data, including 48 from single ARTI children and 164 from multiple ARTIs children, were subjected to cytokine profile measurement. Twenty healthy children who received physical examination at the same hospital, and having matched age and gender with ARTI children, were recruited as non-ARTI control group (Supplementary Table 1). Five multiple-ARTI samples were excluded due to poor quality of cytokine assay, and thus results from 207 ARTI samples were presented (Fig. 1). A total of 48 cytokines or chemonkines were measured using the proteomic chip-based assay (Raybiotech, Norcross GA, USA). These include interleukin-1α (IL-1α), IL-1β, IL-1Rα, IL-2, IL-4, IL-5, IL-6, IL-6 R, IL-7, IL-8, IL-9, IL-10, IL-11, IL-12 p40, IL-12 p70, IL-13, IL-15, IL-16, IL-17A, IL-23, B cell-attracting chemokine (BLC)-1, Eotaxin-1, Eotaxin-2, Fractalkine, granulocyte colony-stimulating factor (G-CSF), granulocyte-macrophagecolony-stimulating factor (GM-CSF), macrophagecolony-stimulating factor (M-CSF), intercellular cell adhesion molecule (ICAM)-1, I-309, interferon (IFN)-induced protein (IP-10), monocyte chemotactic protein-1 (MCP-1), monokine induced by gamma interferon (MIG), macrophage inflammatory protein (MIP)-1α, MIP-1β, MIP-1δ, platelet-derived growth factor BB (PDGF-BB), RANTES, tissue inhibitor of metalloproteinases (TIMP)-1, TIMP-2, tumor necrosisfactor (TNF)-α,TNF-β, TNF RI, TNF RII, transforming growth factor (TGF)-β1, IFN-β, IFN-γ, vascular endothelial growth factor (VEGF), and fibroblast growth factor basic (bFGF). Assays were performed according to themanufacturer's instructions (Raybiotech). Each sample was measured in quadruplicates. Data were automatically processed by using Raybiotech Q-Analyzer Tool with individual standard curve produced from each cytokine standard. Signals were normalized using internal, positive and negative controls included on the array.

**Statistical analyses**. The Shannon Diversity Index, which measures species diversity and relative abundance, was calculated by $H = -\Sigma_{(pi} \log_{(pi)})$. Chao Richness Score, which computes the number of species in a community, was calculated by manual enumeration. Both Shannon diversity and Chao richness were measured at species level. The heat-map of the respiratory virome and phage abundance was obtained using the Origin Pro 2017. Differences of Shannon Diversity and Richness among groups, and the difference of the continuous variables (cytokine levels) between groups were determined by Mann–Whitney U test or nonparametric Kruskal-Wallis Test, correcting for multiple comparisons with Dunn's procedure. A difference with $P < 0.05$ was considered to be statistically significant. Relative abundance of bacteriophages was calculated by the proportion of the normalized reads number[28]. Two sided Fisher's exact test was used to compare the categorical variables (proportions between groups) with a pre-specified significance level of 0.05. Serum levels below detection limits were designated a value zero for the purpose of statistical analysis. A stepwise logistic regression model with backward elimination was used to determine the predictors of recurrent ARTIs. All cytokines and other covariates, such as age, gender, sampling time and season, were analyzed using univariate analysis. Variables with a significance level of less than 0.1 in the univariate analysis were subjected to the logistic regression analysis. The serum levels of all cytokines were log2- and Z-score-transformed and then analyzed by logistic regression analyses with backward elimination. ROC curve analysis was performed to determine the sensitivity and specificity of different variables for prediction of susceptibility to multiple ARTIs. All statistical tests were performed using the SPSS19.0, R package vegan 2.4-4 and Graphpad Prism 6.0.

**Reporting Summary**. Further information on research design is available in the Nature Research Reporting Summary linked to this article.

## Data availability

All the sequencing data excluding human sequences were deposited in the CNSA (https://db.cngb.org/cnsa/) of CNGBdb (project number CNP0000429) under the accession number CNS0092588-CNS0092799. All the software used in this study are available from open source.

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

## Acknowledgements

We thank Dr. Huajie Yan and other clinicians at Shanghai Nanxiang Hospital for their help in sample collection, Profs. Qibin Leng and Xiao Su at IPS, CAS, for their helpful suggestions on the cytokine analysis, Mr. Kai Liu at IPS, CAS, Prof. Ailin Tao and Dr. Yuyi Huang at Guangzhou Medical University for their help in bioinformatics analysis, Dr. Béatrice Regnault at Institut Pasteur (Paris) for her kind help in data interpretation for respiratory virome, and Dr. Beibei Jiang at Max Planck Institute of Psychiatry for her suggestion on statistical analyses. This work was supported by grants from the National Science and Technology Major Project of China (2017ZX10103009–002), the Strategic Priority Research Program (XDB29010000) and the One Belt One Road project (153831KYSB20170043) of Chinese Academy of Sciences, the 133 project of Institut Pasteur of Shanghai, CAS, and Shanghai Sailing Program (16YF1412500).

## Author contributions

C.Z. conceived the study idea. C.Z., Y.L. and J.H.W. designed this study. W.D. and K.L. collected clinical samples and data. Y.L. and X.F. performed sample processing and amplification experiments. J.M. and Q.L. performed the metagenomic sequencing and analysis. Y.L., X.F., C.Z., J.H.W., X.J., J.Z., Y.H. and Z.W. analyzed the data. Y.L., C.Z., J.H.W., X.J., K.L. and Y.Q.K. interpreted the data. Y.L. and C.Z. wrote the first draft of the manuscript. J.X. contributed to critical revision. All authors contributed to the final manuscript.

## Additional information

**Competing interests:** The authors declare no competing interests.

