## [Peer Review File · Nature Communications]

Reviewers' Comments:

Reviewer #1:

Remarks to the Author:

The authors describe a study in which they have used sequencing analysis to identify the virome associated with ARTIs. They were looking for biomarkers that indicate which children might be more susceptible to multiple ARTIs, and they used the sequencing data and some cytokine analysis to look for those markers. They posit that Propionibacterium phage and several cytokines are biomarkers of ARTI and posit that the data suggest antibiotic treatment may be appropriate in these cases. While the data presented are of interest in terms of a longitudinal study of viral infections, there are limitations in the study design, statistical analysis, and conclusions that would need to be addressed before publication.

Major points:

1. I have concerns about the classification of patients based on the number of ARTIs. How is it possible that every ARTI would have been captured in the data collection? Not every ARTI would require a visit to a clinic or hospital.
2. What are the ages of the children in each group? This should be analyzed as a covariate in addition to the tables. There is a trend toward higher age in the multiple ARTI group.
3. In what seasons were the samples collected from each group? Is this matched between groups? Seasonality should be considered as a covariate.
4. Antibiotic usage for other illnesses and antibiotic treatments of the multiple ARTI group should be considered covariates.
5. Line 124 – The Shannon diversity and Chao richness calculations, as I understand them, were done not only using human viruses but also phage. Not only pathogens but potentially non-pathogenic viruses. Therefore, I don't think that this supports virus species "attacking" the airways over time.
6. The time between ARTIs and sampling should be considered as a covariate.
7. How were anelloviruses typed from short-read sequencing? 39 types of anelloviruses at what taxonomic level? Need citations to support that anelloviruses are not pathogenic in the respiratory tract.
8. Why were the authors unable to assess the respiratory bacteria? Be specific (line 138)
9. Line 145 says the diversity were consistent with abundance. Was this sequencing done in a way to make it quantitative? If so, specify in methods. Otherwise remove reference(s) to quantitative results.
10. Line 145: "The above results imply that bacteria and viruses played differential roles in recurrent ARTIs." This is discussion, not results. Other sentences throughout results should also be moved to discussion. What is the support for this statement? No roles have been defined and no causation demonstrated, so it is unclear how these bacteriophage profiles indicate differential roles for bacteria and viruses.
11. Line 171 – this paragraph says it is exploring mechanism, but it is not. There is no link demonstrated between the phages and the cytokines that are assayed.

12. Line 203 – there has been no link shown between Propi phage "acquisition" and increased serum cytokine levels.

13. Line 233. While the association between Propi phages and recurrent ARTIs is interesting, and there is likely a biological explanation that lies in viral and bacterial community dynamics, these data in no way support the idea that antibiotics could be or should be applied to these cases. It is highly probable that the dynamics of the bacteria and viral communities need to be assessed together and considered in context of treatments. It would be important to understand whether P. phage presence truly associates with MORE Propionibacterium, and whether an increase in signal from of Propi or Propi phage results from a loss of other bacteria or an overgrowth of Propi.

14. I am a little confused about how the Multiple ARTIs are presented in table 1, as I expect that each episode probably had different symptoms. I think these should be broken down by episode. Those with diagnosis of tonsillitis, pharyngitis, bronchiolitis and pneumonia should be added. Were those diagnoses more likely to fall into one group or the other? And at which visit if in the multi ARTI group?

15. Methods: Line 391 "After ligation" – ligation of what?

16. Phi29 would NOT be quantitative and will skew your results for small circular viruses. This should be included as a limitation of the study. Small circular viruses include anelloviruses. Some Propionibacterium phage are also circular. Could you have selected for these particular viruses? Are the genomes you amplified circular?

17. Is there a citation for the BGI sequencing method? DNA nanoballs?

18. Line 415 – type SURPI. Please clearly highlight what changes were made from the SURPI pipeline and in what mode you were running the pipeline. Please add parameters for the software programs used.

19. What were translated sequences aligned with?

20. What were contigs aligned with? If NCBI Blastx, needs to be clarified and cited.

21. How were reads taxonomically classified?

22. Why was the arbitrary cut off of 15 reads chosen? If you do not use that cutoff, do you still get the same associations? This is very important.

23. Figures: no letters on the panels in Figure 1

24. Figure 1 Should not combine the data from the multiple ARTIs

25. Why does figure 2 e show bacteria if bacteria could not be assessed based on methodological limitations?

26. Figure 2 b – I am not confident in the assessment of anellovirus types. Some of these taxa have higher resolution than others. Were they detected by nucleotide alignment or translated alignment? Translated would limit classification to a specific type, and you would have to be cautious to classify to

a subtype with nucleotide alignments if you used short reads. Were reads or contigs used here? If contigs, was there a problem with contig assembly with similar viruses?

27. Minor points:

Abstract:

- Define ARTIs in parentheses after it is spelled out in the first sentence.
- "multiple ARTIs" is a little unclear. Might say "multiple ARTIs over time" or something to that effect.
- Line 52: "... increased in patients with multiple ARTIs over time compared with patients with a single ARTI and non-ARTI controls...."
- Line 54: "Propionibacterium phages were"
- Line 58-61: These findings do not suggest antibiotics as a treatment for recurrent ARTIs.
- Line 153 – mentions "two groups" – which two groups? There are 3 groups of anelloviruses.

Reviewer #2:

Remarks to the Author:

Title: Elevated respiratory Propionibacterium phage abundance and serum TIMP-1 and PDGF-BB levels are novel biomarkers of susceptibility for recurrent acute respiratory tract infections in children.

Summary findings: The aim of this study was to identify novel biomarkers of susceptibility to frequent ARTIs in children. Three potential biomarkers of multiple-ARTIs were identified, i) abundant Propionibacterium phage in the upper respiratory tract (URT) and elevated serum levels of ii) TIMP-1 and iii) PDGF-BB. Taken together, these markers achieved an AUC score of 0.94 on ROC analysis suggesting a strong predictive value. This study presents novel findings that have potential clinical benefit. However, the authors should address the concerns listed below.

Reviewer comments:

- A careful review of the manuscript's English grammar is recommended. For example (lines 343-44), "Alternatively, having a clinical diagnosis of tonsillitis, pharyngitis, bronchiolitis and pneumonia." is not a complete sentence.
- Abstract, line 57. Spell out "AUC".
- Methods, study design, lines 336-351. As acknowledged by the authors, this study is flawed in several important ways. First, by its retrospective design. Children in the single-ARTI group may have experienced multiple ARTIs occurring before enrollment or after study completion. Moreover, ARTI episodes are only captured if children attend the study clinic. Children with mild ARTIs that would not trigger a clinic visit, visits to clinics outside of the study catchment area, or children that have moved out of the catchment area would not be captured. Data on asthma and other chronic respiratory conditions were not collected that might account for multiple ARTIs in some study children. Finally, opportunities for infection are an important factor determining multiple-ARTI that the authors appear to have not considered. Potential for infection depends upon what respiratory viruses are circulating in the community which can vary by year and season. Children attending daycare, school or living in households with multiple siblings would be more susceptible to multiple respiratory virus infections, independent of underlying host susceptibility.
- Methods, study design, lines 346-351. Details on clinical specimen collection and storage should be

provided. The authors note that, "children did not receive clinical treatment including antibiotics treatment when samples were collected". How did the authors account for potential self-medication with non-prescribed antibiotics?

- Methods, proteomic chip-based cytokine assay, lines 446-447. Given that acute serum specimens were available for cytokine analysis, did the authors consider adding serologic testing as an indicator of immune status to at least the major respiratory viruses? NGS or PCR analysis of the sera for respiratory viruses/phage might also provide insight into the integrity of airway mucosa.

- Methods, proteomic chip-based cytokine assay, line 448-449. Were the 20 asymptomatic control children matched for age and gender?

- Methods, selection criteria ... , lines 366, 369, 410, Table 1. For added clarity, Table 1 should note that 2 of the 50 single ARTI and one of the 62 multiple ARTI children were excluded from the analysis (referral to Table 1 might be added to line 411).

- Methods, nucleic extraction ... , lines 437-443. Please clarify the purpose of testing for HRV, HEV, IFV and RSV using specific RT-PCR assays. Were these assays used to determine the "% of samples positive for common RVBs" in Figure 2c?

- Results, Line 117. How do the authors explain their failure to identify human bocavirus, which would be expected to be found in this age group?

- Results, Line 117 & Fig. S2. The 34 virus species identified should not be co-classified as "pathogenic respiratory viruses". Although found in the respiratory tract, a suggested clinical classification scheme follows:

1. Respiratory viruses with low proportions of subclinical infections

- a. Influenza A & B
- b. Human respiratory syncytial virus
- c. Human parainfluenza viruses 1-4
- d. Human metapneumovirus
- e. Human mastadenovirus B

2. Respiratory viruses with high proportions of subclinical infections

- a. Rhinovirus A, B, C
- b. Enterovirus A, B, D, sp.
- c. Influenza C
- d. Human mastadenovirus C
- e. Human coronaviruses OC43, NL63, 229E

3. Viruses rarely etiologically associated with ARTIs

- a. Human herpes viruses
- b. Polyomaviruses

4. Viruses not etiologically associated with ARTIs

- a. Adeno-associated viruses
- b. Papillomaviruses

- Results, lines 119-124, Fig 1a. Despite a reasonable ≥ 30 day waiting period between consecutive episodes, single ARTI infections with a definitive respiratory virus pathogen (see above) might trigger downstream reactivation and shedding of DNA viruses that evidence persistent infections and may

account for increases in co-detections on subsequent ARTIs. These viruses include herpes viruses, polyomaviruses, papillomaviruses, species C adenoviruses and adeno-associated virus. The authors should consider removing these viruses from their analysis to see if their findings hold.

- Results, lines 138-139. Can the authors provide a reference showing that bacteriophage “profiles” in the URT correspond quantitatively with their respective bacteria?
- Results, line 156. Do the authors mean “commensal” instead of “communal”?
- Results, lines 162-163. Is there a reason that the authors chose Lactococcus phage as a comparator with Propionibacterim phage from among the available options? It is interesting to note that Erwinia phage show a significant but reciprocal difference in levels between single- and multiple-ARTI children and show a progressive decrease with ARTI episodes (Figure S3). By the same logic, is this a potential biomarker for multiple-ARTI?
- Discussion, lines 233-235. The authors should be cautious in interpreting their finding of a quantitative correlation of Propionibacterium phage levels in the URT with multiple ARTIs. Correlation does not equal causation.
- Discussion, lines 279-290. Can the authors cite any reference that Propionibacterium phage or their host bacteria are potentially pathogenic in the URT of otherwise healthy children? If elevated levels of these bacteria in the URT represent a dysbiosis of the URT that could lead to colonization with definitive bacterial pathogens, why not just test for these pathogens directly (*S. pneumoniae*, *H. influenza*, *S. aureus*, *S. pyogenes*, *M. catarrhalis*) using specific PCR assays to evaluate this hypothesis?
- Discussion, lines 273-277. That increased levels of TIMP-1 and PDGF-BB have been found in patients with asthma/COPD, and that these chronic conditions could account for multiple-ARTI episodes (exacerbations) in these study patients, it is essential to ascertain this information as part of the study design.
- Discussion. Are there practical/cost limitations to using both undirected NGS to assess the URT virome and cytokine analysis of serum as a clinical laboratory test for multiple ARTI susceptibility? Given that a child is identified as potentially susceptible to multiple ARTIs, what practical interventions would the authors suggest that would help reduce this risk?

A point-by-point response to the reviews' comments

Reviewer #1 (Remarks to the Author):

The authors describe a study in which they have used sequencing analysis to identify the virome associated with ARTIs. They were looking for biomarkers that indicate which children might be more susceptible to multiple ARTIs, and they used the sequencing data and some cytokine analysis to look for those markers. They posit that *Propionibacterium* phage and several cytokines are biomarkers of ARTI and posit that the data suggest antibiotic treatment may be appropriate in these cases. While the data presented are of interest in terms of a longitudinal study of viral infections, there are limitations in the study design, statistical analysis, and conclusions that would need to be addressed before publication.

Major points:

1. I have concerns about the classification of patients based on the number of ARTIs. How is it possible that every ARTI would have been captured in the data collection? Not every ARTI would require a visit to a clinic or hospital.

Response: We agree with the reviewer that classification of patients is critical for the interpretation of our results. Despite unable to capture all patients, we are confident in having captured most patients, because of the uniqueness of Chinese local health care system. Specifically, in this study, we collected data from Nanxiang hospital, the only general hospital in Jiading District of Shanghai with a strong pediatric practice and being the referral centre for clinically ill children. Granted some children with ARTI episodes might have been missed due to several reasons including but not limited to “patients went to different hospitals, or self-medication at home” as we stated in the revised manuscript. Similar to many clinical studies, we believe that with the appropriate sample size, the fundamental observation should still reflect some truth. Indeed, significantly elevated serum TIMP-1 and PDGF-BB levels and respiratory *Propionibacterium* phage abundance were observed in the multiple-ARTIs children in comparison to the single-ARTI group. The above thought process in relation to the limitation of our study has been discussed in the revised Discussion section.

2. What are the ages of the children in each group? This should be analyzed as a covariate in addition to the tables. There is a trend toward higher age in the multiple ARTI group.

3. In what seasons were the samples collected from each group? Is this matched between groups? Seasonality should be considered as a covariate.

Response to Q2-3: To address these questions, we modified Table 1 in which the reviewer concerned covariates were highlighted. To be specific, the median ages of the single-ARTI group and the first episode of the multiple-ARTIs group were 40.8

and 45.6 ($p=0.082$), respectively. The ages for subsequent episodes were not taken into account because children's ages have unquestionably become older when they experienced more than one episode of infection during the study period. The age of the first episode of the multiple-ARTIs group was slightly higher, but not statistically significant ($p=0.082$). We also added the season distribution data into the Table 1. As expected, the seasonal distribution of single ARTI and multiple ARTIs was not statistically different (Table 1).

As suggested, we added the age and the season, as well as other factors (e.g. the time between ARTIs and sampling), as covariates into consideration for logistic regression analyses. Because there was an inevitable increase in the ages at later episodes, we used the ages at the first episode of the multiple-ARTIs group (same procedure was also used in *J Allergy Clin Immunol.* 2016. 137(3):774-81), or the median of ages of each children with multiple episodes over time into the analyses. Moreover, all the variables were normalized by “Log” and “Z-score” to reduce the effects of data dispersion of different variables. Our new analyses showed that the seasonal factor was not significantly associated with the occurrence of multiple ARTIs, but we observed a weak association between age and the occurrence of multiple ARTIs ($P=0.034-0.039$). However, the adjusted ORs (1.56-1.51) for the age factor were substantially lower than those of cytokines TIMP-1 and PDGF-BB (Table 2). Having included the reviewers suggested analysis of covariates, our conclusions remain the same, in that all four logistic regression analyses verified that two cytokines TIMP-1 (P values ≤ 0.001 ; adjusted ORs: 5.05-61.56) and PDGF-BB (P values: 0.078-0.003; adjusted ORs: 2-11.58) were significantly associated with the occurrence of multiple ARTIs (Table 2 and Tables S2-S4). Therefore, significantly elevated serum levels of TIMP-1 and PDGF-BB might be potential indicators for the susceptible/immune status of airway to respiratory virus infection.

“Table 2 in Main text” Risk factors associated with multiple ARTIs (first episode) using logistic regression (Log2 transformed).

Predictive variables	P value	Adjusted OR	95% CI	
			Lower	Upper
Log2 TIMP-1	<0.001	61.56	8.19	462.87
Log2 PDGF-BB	0.005	11.58	2.08	64.61
Age	0.039	1.51	1.02	2.24
Log2 ICAM-1	0.078	1.57	0.95	2.61

“Table S2 in Supplementary materials” Risk factors associated with multiple ARTIs (first episode) using logistic regression (Z-score transformed).

Predictive variables	P value	Adjusted OR	95% CI	
			Lower	Upper
Z-score TIMP-1	<0.001	5.05	2.24	11.40
Z-score PDGF-BB	0.005	3.04	1.41	6.57
Z-score Eotaxin-1	0.105	2.06	0.86	4.92
Z-score ICAM-1	0.087	1.80	0.92	3.54
Age	0.034	1.56	1.03	2.37
Z-score TNFR11	0.100	0.57	0.29	1.11

“Table S3 in Supplementary materials” Risk factors associated with multiple ARTIs (median of different episodes) using logistic regression (Log2 transformed).

Predictive variables	P value	Adjusted OR	95% CI	
			Lower	Upper
Log2 TIMP-1	0.001	50.65	5.33	481.33
Log2 PDGF-BB	0.003	11.58	3.14	299.33
Log2 IL-17	0.124	0.75	0.52	1.08
Log2 IL-15	0.087	0.53	0.26	1.10
Log2 IL-6	0.018	0.42	0.20	0.86

“Table S4 in Supplementary materials” Risk factors associated with multiple ARTIs (median of different episodes) using logistic regression (Z-score transformed).

Predictive variables	P value	Adjusted OR	95% CI	
			Lower	Upper
Z-score TIMP-1	<0.001	6.02	2.47	14.71
Z-score PDGF-BB	0.078	2.00	0.93	4.31
Z-score IL-6	0.119	0.58	0.30	1.15
Z-score IFN- β	0.038	0.51	0.27	0.93
Z-score IL-17	0.093	0.54	0.26	1.11
Z-score TNFR11	0.100	0.57	0.29	1.11

4. Antibiotic usage for other illnesses and antibiotic treatments of the multiple ARTI group should be considered covariates.

Response: We agree with the reviewer that antibiotic usage is a crucial factor to be considered as a covariate in the analysis of ARTIs. As far as we know, all clinical samples included in this study were collected prior to antibiotics treatment. Furthermore, as a prescription drug, antibiotics are not available to the general public without medical record, especially in Shanghai where the rules and regulations are strictly followed. So, the possibility of self-medication for the children with ARTIs before they visited the hospital was small. Nonetheless, the concern is legitimate, and it was described in Methods.

5. Line 124 – The Shannon diversity and Chao richness calculations, as I understand them, were done not only using human viruses but also phage. Not only pathogens but potentially non-pathogenic viruses. Therefore, I don't think that this supports virus species "attacking" the airways over time.

Response: We appreciated the reviewer's concern which is shared by another reviewer. To address these comments, we performed additional analyses using the data of pathogenic respiratory viruses by removing non-pathogenic or non-ARTI related viruses (i.e. herpes viruses, polyomaviruses, papillomaviruses, and adeno-associated virus). Results showed that a similar trend of higher Shannon diversity and Chao richness in the multiple-ARTIs group than the single-ARTI group was also observed when these viruses were removed (Figure 1b, c, e and f in the revised manuscript). These additional analyses are also in supportive of our original conclusion.

6. The time between ARTIs and sampling should be considered as a covariate.

Response: We added the data of time between ARTIs and sampling into the Table 1, and re-performed logistic regression analyses including the time between ARTIs and sampling. The results showed that time between ARTIs and sampling was not significantly associated with the occurrence of multiple ARTIs. (Please see the Methods, "Table 2" and "Tables S4-6" in the revised manuscript, and the answer to Q2-3).

7. How were anelloviruses typed from short-read sequencing? 39 types of anelloviruses at what taxonomic level? Need citations to support that anelloviruses are not pathogenic in the respiratory tract.

Response: Anelloviruses were classified at species level (see the Methods of the revised manuscript). Some previous studies reported that anelloviruses are widely found among humans, including both healthy and sick people (*PLoS Pathog* 2017, 13(3):e1006292; *Nat Med.* 2015, 21(10):1228-34; *Rev Med Virol* 1999, 9:73–74; *Virology* 2009, 6:134; *J Clin Microbiol* 2008, 46:507–514, etc.). Currently, there is no evidence to support that anelloviruses are specifically associated with certain human disease. We added these references in support of our previous statement.

8. Why were the authors unable to assess the respiratory bacteria? Be specific (line 138)

Response: The main goal of this study was designed to investigate the virome associated with ARTIs in Children; therefore, experimental methods of NGS were not optimized for detection of bacteria. Despite this, we also tried to determine the

respiratory microbiota by v3v4 amplification and NGS. The effort was unsuccessful, possibly due to small sample quality, freezing and thawing of samples, and/or low abundance of bacteria in the samples (a swab in 3ml medium). We mentioned this as a limitation in the manuscript.

9. Line 145 says the diversity were consistent with abundance. Was this sequencing done in a way to make it quantitative? If so, specify in methods. Otherwise remove reference(s) to quantitative results.

Response: We used both NGS and metagenomic analysis to classify taxonomy of viral species, as well as quantifying the abundance of these viruses. The abundance was estimated by normalizing the reads number as previously described (*Cell Host & Microbe* 2016, 19: 311–322; *Cell Host & Microbe* 2016, 19: 323–335). It was described in the Methods section. In addition, the results in Figure S3 (Previous Fig. 1g-i) was not involved in abundance; so we revised the sentence into “The Shannon diversity and Chao richness of bacteriophages had consistent trends and were not significantly different between the two patient groups”.

10. Line 145: "The above results imply that bacteria and viruses played differential roles in recurrent ARTIs." This is discussion, not results. Other sentences throughout results should also be moved to discussion. What is the support for this statement? No roles have been defined and no causation demonstrated, so it is unclear how these bacteriophage profiles indicate differential roles for bacteria and viruses.

Response: We agree with the reviewer’s comments. As suggested, we deleted this sentence and removed the discussions from Results to Discussion. In addition, we also reorganized the results in Figure 1 by removing the original panels 1h-g to the supplementary Figure S3.

11. Line 171 – this paragraph says it is exploring mechanism, but it is not. There is no link demonstrated between the phages and the cytokines that are assayed.

Response: We agree that our study is not "mechanistic" by conventional definition, and thus changed the text accordingly.

12. Line 203 – there has been no link shown between Propi phage "acquisition" and increased serum cytokine levels.

Response: We agree that "link" is a stronger statement than our data can justify. In fact, we showed that the presence of *Propionibacterium* phage was associated with higher TIMP-1/PDGF-BB levels in the Figure 4c-f, and the presence or absence of any other viruses did not significantly affect the association between the presence of

Propionibacterium phage and higher TIMP-1/PDGF-BB levels in Figures S7-S8. Therefore, we changed "acquisition" to "presence".

13. Line 233. While the association between Propi phages and recurrent ARTIs is interesting, and there is likely a biological explanation that lies in viral and bacterial community dynamics, these data in no way support the idea that antibiotics could be or should be applied to these cases. It is highly probable that the dynamics of the bacteria and viral communities need to be assessed together and considered in context of treatments. It would be important to understand whether P. phage presence truly associates with MORE Propionibacterium, and whether an increase in signal from of Propi or Propi phage results from a loss of other bacteria or an overgrowth of Propi.

Response: We agree with the reviewer that there was no data to support that antibiotics could be or should be applied to these cases. So, we deleted the sentence "The identification of *Propionibacterium* phages ... could lead to inclusion of antibiotics treatment into the clinical management guidelines" in the revised manuscript. With respect to the dynamics of the bacterial and viral communities, we gave some discussions in the Discussion section.

It is clear that phages correlate with their bacterial hosts. Phages typically outnumber their bacterial hosts and most of the time there is a balance between phages and their hosts (*Adv Appl Microbiol.* 2014, 89:135-83, *Nat Immunol.* 2013, 14(7):654-9; *Cell* 2015, 160, 447–460). The interaction between phages and bacteria is very complex. Phages can not only decrease the abundance of specific bacteria, and also increase their abundance by conferring advantage over other bacterial community members. Lytic phages may have a negative correlation with their host population density, whereas there may be a positive correlation for temperate phages.

To determine the correlation between *Propionibacterium* phages and *Propionibacterium*, as well as between other phages (e.g. *Lactococcus* phages, *Enterobacter* phages, *Klebsiella* phages, *Pseudomonas* phages, *Streptococcus* phages and *Escherichia* phages) with high prevalence in throat swab samples and their hosts, we performed linear-regression analyses (spearman correlation) using the phage reads and bacteria reads abundance obtained from NGS. The results showed that most the top phages (*Propionibacterium* phages, *Lactococcus* phages, *Enterobacter* phages, *Klebsiella* phages and *Pseudomonas* phages) in this study had a significant but weak positive correlation with their bacterial hosts, while no significant correlation was detected for *Streptococcus* (Figure S5) and *Escherichia* phages (data not shown) and their hosts. These results suggest that the presence of *Propionibacterium* phages associates with relatively more *Propionibacterium*, and decreased abundance of *Lactococcus* phages and other phages imply decreased density of *Lactococcus* and other corresponding bacteria. We revised the corresponding text, and provided

necessary references in the revised manuscript. In addition, we deleted the sentence about the antibiotics usage as potential treatment in the revised manuscript.

Figure S5 in Supplementary materials. The correlations of six top phages with their bacterial hosts. The result of Escherichia phages and their hosts was not shown.

14. I am a little confused about how the Multiple ARTIs are presented in table 1, as I expect that each episode probably had different symptoms. I think these should be broken down by episode. Those with diagnosis of tonsillitis, pharyngitis, bronchiolitis and pneumonia should be added. Were those diagnoses more likely to fall into one group or the other? And at which visit if in the multi ARTI group?

Response: As suggested, the demographic and clinical characteristics of the multiple-ARTIs group were broken down by episode in the Table 1. Because there is

only one case of five episodes, the data at the fifth episode was shown together in the fourth episode. In addition, the information on the sampling date, season, and clinical diagnosis (tonsillitis, pharyngitis, bronchiolitis and pneumonia) were added. We only presented the clinical diagnosis information of about half of the cases, because the rest was not available. According the limited diagnosis data, the majority of the cases was diagnosed as tonsillitis (35-68.9%) and bronchitis (27.6-50%), and there was no significant different among different groups (Table 1).

15. Methods: Line 391 "After ligation" – ligation of what?

16. Phi29 would NOT be quantitative and will skew your results for small circular viruses. This should be included as a limitation of the study. Small circular viruses include anelloviruses. Some Propionibacterium phage are also circular. Could you have selected for these particular viruses? Are the genomes you amplified circular?

Response to Q15-16: In the study, the unbiased multiple displacement amplification (MDA) was used for the enrichment of viral sequences (REPLI-g Single Cell WTA kit, Qiagen, Germany). Because of good amplification efficiency, MDA was widely used for virome study. Ligation is a recommended step by the kit. The reason is that MDA uses large-fragments of DNA as templates for random amplification by phi29 DNA polymerase, thereby recommending first to obtain large-fragments of DNA by ligation of small DNA fragments. A detailed methodological description was added in the Methods section of the revised manuscript.

Phi29 DNA polymerase that has strong strand displacement activity was widely used in the enrichment of low-quantity templates for NGS (*Cell Host Microbe*. 2016, 19 (3):311-22; *Proc Natl Acad Sci U S A*. 2017, 114(30):E6166-E6175; *Nature*. 2010, 466 (7304):334-8). It can generate high NGS coverage (*Gigascience*. 2015, 4:37; *Annu. Rev. Genomics Hum. Genet.* 2015, 16:79–102), but also lead to preferential amplification of circular DNA. In this study, the use of MDA did not specially focus on circular viruses, and the initial large-fragments of DNA templates used for subsequent NGS were not circular. In the analyses, we divided the respiratory virome into three panels, common respiratory viruses, anelloviruses, and bacteriophages. Among bacteriophages, microviridae, a family of bacteriophages that has a single-stranded circular DNA genome, was not preferentially amplified (Figure S2). Therefore, we think that the MDA was less likely to selectively skew our results for circular *Propionibacterium* phages. In view of the fact that some *Propionibacterium* phages are indeed circular, we mentioned this as a limitation in Discussion.

17. Is there a citation for the BGI sequencing method? DNA nanoballs?

Response: The BGI sequencing platform BGISEQ-500 uses a variation on the “DNA nanoballs” sequencing method. We added two references (Comparative performance

of the BGISEQ-500 vs. Illumina HiSeq2500 sequencing platforms for palaeogenomic sequencing. *Gigascience*. 2017, 6(8):1-13 and Human genome sequencing using unchained base reads on self-assembling DNA nanoarrays. *Science* 2010, 327 (5961): 78–81) for the BGI sequencing method and platform in the revised manuscript.

18. Line 415 – type SURPI. Please clearly highlight what changes were made from the SURPI pipeline and in what mode you were running the pipeline. Please add parameters for the software programs used.

Response: We added the full name (sequence-based ultra-rapid pathogen identification) of SURPI when first mentioned in the text. In this study, we used an in-house pipeline based on the SUPPI’s principle to identify viruses. Similar to the comprehensive mode of SURPI, the in-house pipeline includes quality control (preprocessing), removing of host sequence, classification, verification and filtering. To remove host sequence, we used more human sequences databases including a Chinese ethnicity. In the step of classification, we used KRAKEN instead of SNAP to map against the reference databases. For verification, we first blasted the viral reads to the NCBI non-redundant NT database. After removing false positive viral hit, the remaining reads were assembled and blasted against the NCBI NT database to achieve more accurate classification. Because a small number of reads detected from some samples can be the “bleed over” from other samples on the flow cell or environment (*Nat Commun*. 2018. 9(1):4270; *PLoS One* 2015. 10:e0120520; *PLoS Pathog* 2017.13:e1006292), thereby representing false positives, in the step of filtering, only the reads matching to virus references exceeded 15 was considered as positive to reduce the potential false positives (*Nat Med* 2017. 23:1080-1085). We added a detailed methodological description and provided related references in Methods section of the revised manuscript.

19. What were translated sequences aligned with?

Response: We apologize for this mistake in methodology description. We have corrected it in the revised Methods.

20. What were contigs aligned with? If NCBI Blastx, needs to be clarified and cited.

Response: The contigs and unassembled reads were aligned with the NCBI NT database. We have revised the methods to clarify this and cited a reference.

21.How were reads taxonomically classified?

Response: The reads were classified by the software KRAKEN and a related reference was cited in the revised manuscript (please also see the answer to Q18).

22. Why was the arbitrary cut off of 15 reads chosen? If you do not use that cutoff, do you still get the same associations? This is very important.

Response: False positive caused by “bleed over” is a common problem faced by viral metagenomic analysis (*Nat Commun.* 2018, 9(1):4270). In the *Nat Commun* paper, the authors used 0.4% as a threshold for “bleed over” to reduce false positives. In another study of human virome (*Nat Med.* 2017, 23 (9):1080-1085), 15 reads were used as the cut off for a positive result. In our study we chose a cut off of 15 reads, which is same to the cutoff used in the *Nat Med* paper. Furthermore, we measured the taxa distribution at different cutoff levels from 1 to 50 (Figure R1). We found that the number of taxa identified remained stable when the cutoff were ≥ 15 reads (Figure R1). Therefore, we believed that the cutoff of 15 reads was more rigorous to avoid false positive results. Furthermore, for the top eukaryotic viruses identified in this study, further confirmation was also performed by qPCR (Figure 2). The related description and references for the choice of cutoff were added in Methods of the revised manuscript.

One of the most important findings in this study was that higher abundance of *Propionibacterium* phages in children with multiple ARTIs over time than in the single-ARTI children, and the reverse for *Lactococcus* phages and other phages. The cutoff influences some of the judgments of positive and negative, but does not change the relative abundance of each phage in the single-ARTI and the multiple-ARTIs groups. To exclude the potential influence of the cutoff value on the results, we performed similar analyses to Figure 3b without a cutoff (i.e. a cutoff of 0, which means one or more reads representing a positive result). We found that although there was a little change in the detection rates of some phages, the distribution trends of the top 15 phages between the single-ARTI and multiple-ARTIs groups were completely consistent with those determined with a cutoff of 15 reads (Figure 3b and Figure R2).

In addition, we also re-performed the analyses presented in the Figure 4c-f. The results showed that TIMP-1 and PDGF-BB levels are still higher in *Propionibacterium* phages-positive group than in the negative group when no cutoff was used (Figure R3). These results indicated that the cutoff levels did not influence the main results and conclusions of this study.

Figure R1. Taxa identified with different cutoff of reads (different colors represent different sequencing batch).

Figure R2 Comparison of the relative abundance of each bacteriophage taxa between children with single and multiple ARTIs using Mann-Whitney U test. (Corresponding to the Figure 3b in the revised version)

Figure R3. Comparison of TIMP-1 and PDGF-BB levels between children who were positive and negative for *Propionibacterium* phage was made by using Mann-Whitney U test (Corresponding to Figure 4c-d in the revised manuscript).

23. Figures: no letters on the panels in Figure 1

Response: We thank the reviewer for pointing out this. The missing information was added in the revised manuscript.

24. Figure 1 Should not combine the data from the multiple ARTIs

Response: In fact, the combined data of multiple ARTIs were shown for comparison with the single-ARTI group. Besides, we also showed the data of each episode of the multiple-ARTIs group in Figure 1.

25. Why does figure 2e show bacteria if bacteria could not be assessed based on

methodological limitations?

Response: We apologized for this error. The data shown in the Figure 3a (original Figure 2e) were bacteriophage taxa (described in the figure legend), not bacteria taxa. We have corrected it in the revised manuscript.

26. Figure 2 b – I am not confident in the assessment of anellovirus types. Some of these taxa have higher resolution than others. Were they detected by nucleotide alignment or translated alignment? Translated would limit classification to a specific type, and you would have to be cautious to classify to a subtype with nucleotide alignments if you used short reads. Were reads or contigs used here? If contigs, was there a problem with contig assembly with similar viruses?

Response: We agree with the reviewer that there was difficulty in contig assembly with similar viruses. To avoid misclassification of virus types, the anellovirus types were identified by nucleotide alignment using the in-house pipeline, and verified by BLAST to the NT database with reads or contigs. All the reads or contigs were carefully checked manually to make sure of correct classification. To identify an anellovirus type, at least 3 reads or contigs should be mapped to only one type (unique map). The reads or contigs that are unable to be mapped to one type of anellovirus were discarded for further classification.

27. Minor points:

Abstract:

-Define ARTIs in parentheses after it is spelled out in the first sentence.

-"multiple ARTIs" is a little unclear. Might say "multiple ARTIs over time" or something to that effect.

-Line 52: "... increased in patients with multiple ARTIs over time compared with patients with a single ARTI and non-ARTI controls..."

-Line 54: "Propionibacterium phages were"

-Line 58-61: These findings to not suggest antibiotics as a treatment for recurrent ARTIs.

-Line 153 – mentions "two groups" – which two groups? There are 3 groups of anelloviruses.

Response: Thank you for your suggestions and corrections. All these points were corrected in the revised manuscript. In particular, we updated multiple ARTIs to multiple ARTIs over time in the revised manuscript, and removed the sentences suggesting antibiotics use in treatment of recurrent ARTIs. In line 153, the two groups were the respiratory viruses and anelloviruses. We corrected two groups into “neither certain respiratory viruses nor certain anelloviruses”.

Reviewer #2 (Remarks to the Author):

Title: Elevated respiratory *Propionibacterium* phage abundance and serum TIMP-1 and PDGF-BB levels are novel biomarkers of susceptibility for recurrent acute respiratory tract infections in children.

Summery findings: The aim of this study was to identify novel biomarkers of susceptibility to frequent ARTIs in children. Three potential biomarkers of multiple-ARTIs were identified, i) abundant *Propionibacterium* phage in the upper respiratory tract (URT) and elevated serum levels of ii) TIMP-1 and iii) PDGF-BB. Taken together, these markers achieved an AUC score of 0.94 on ROC analysis suggesting a strong predictive value. This study presents novel findings that have potential clinical benefit. However, the authors should address the concerns listed below.

Reviewer comments:

1. A careful review of the manuscript's English grammar is recommended. For example (lines 343-44), "Alternatively, having a clinical diagnosis of tonsillitis, pharyngitis, bronchiolitis and pneumonia." is not a complete sentence.

Response: Thank you very much for pointing out some errors in English usage. Other than carefully checked and revised the language of the manuscript ourselves, we enlisted an American citizen who is a professor of virology to edit the manuscript thoroughly. We believed the English of our current version has been greatly improved.

2. Abstract, line 57. Spell out "AUC".

Response: We provided the full name (area under the curve) of "AUC" in the revised version.

3. Methods, study design, lines 336-351. As acknowledged by the authors, this study is flawed in several important ways. First, by its retrospective design. Children in the single-ARTI group may have experienced multiple ARTIs occurring before enrollment or after study completion. Moreover, ARTI episodes are only captured if children attend the study clinic. Children with mild ARTIs that would not trigger a clinic visit, visits to clinics outside of the study catchment area, or children that have moved out of the catchment area would not be captured. Data on asthma and other chronic respiratory conditions were not collected that might account for multiple ARTIs in some study children. Finally, opportunities for infection are an important factor determining multiple-ARTI that the authors appear to have not considered. Potential for infection depends upon what respiratory viruses are circulating in the community which can vary by year and season. Children attending day care, school or

living in households with multiple siblings would be more susceptible to multiple respiratory virus infections, independent of underlying host susceptibility.

Response: We acknowledged that there are some limitations in the retrospective design, especially in the classification of the single-ARTI group. We believed that a few ARTI episodes might have been missed due to reasons raised by the reviewer and ourselves (patients visited to different hospitals, or self-medication for mild ARTIs at home, or moved out of the catchment area). The potential omission might lead to misclassifications of a few cases (who might have two or more ARTI episodes before enrollment or after study completion) in the single-ARTI group, but it is less likely to influence the classification of the multiple-ARTIs group. In spite of this, we still observed significantly elevated serum TIMP-1 and PDGF-BB levels and respiratory *Propionibacterium* phage abundance in the multiple-ARTIs children than the single-ARTI group, implying the effect is strong even if a few cases of multiple-ARTIs children might be misclassified in the single-ARTI group; else, the number misclassified, if any, was small. So, we considered that the multiple ARTIs episodes more likely reflect the susceptibility status of some children for respiratory virus infection, as shown by our results.

We agree that asthma and/or other chronic respiratory conditions influence or contribute to the multiple ARTIs in some recruited children as reflected by increased levels of two inflammatory cytokines TIMP-1 and PDGF-BB in the multiple-ARTIs children, both of which was also increased in asthma and ARDS patients. We reviewed the medical records of the recruited children in this study, and found that only a few of the study patients had these chronic conditions or had asthma history (2 cases in single group, 3 cases in multiple group). The low percentage (2.3%) of asthma in our cohort suggests that asthma and chronic respiratory conditions have limited influence on the occurrence of multiple ARTIs. We added some discussion on this point in the revised manuscript.

To exclude the influence of other factors, we added age, season, gender and sampling time as covariates into our dataset, and re-performed the logistic regression analyses. The results did not show a significant association of any of these added factors with the multiple ARTIs in Children (Table 2 and Table S4-S6) (Please also see the answers to Q2-3 of the first reviewer). On the other hand, according previous studies performed by us and others, influenza virus, RSV, rhinovirus, and parainfluenza viruses were the main respiratory viruses circulating in Shanghai (*Arch Virol.* 2016. 161(7):1907-13; *PLoS One.* 2012. 7(9):e44568; *PLoS One.* 2013. 10(3): e0119513). The top respiratory viruses identified in this study were rhinovirus, influenza virus, enterovirus, CoVs, parainfluenza virus, and RSV, which were consistent with previous observations. Of particular importance was that certain single respiratory virus was not found to be significantly associated with the multiple-ARTIs

children (Figure 2).

In this study, because the information of attending daycare, school or living in households were not available, we were unable to test whether there was an association between the social status of a child and the occurrence of multiple ARTIs over time. However, in China, most children younger than 3 years are living at home, children with 3-6 years old attend kindergarten, and most children older than 6 years attend primary school. We found that the age was not an important factor for the occurrence of multiple ARTIs over time (Table 2 and Table S4-S6), which might suggest that the social status of a child has little contribution to multiple ARTIs.

On the other hand, China implemented the family planning policy (One-child policy) from 1970s to 2016. A young couple, especially those living in big cities, is only allowed to raise one child. In particular, almost all families in Shanghai have one child before 2016. This study recruited children with ARTIs during 2009-2015. Therefore, there was very low possibility of the recruited children having multiple siblings.

Taken all the concerns of the reviewer into consideration, we updated the limitation of our study in the last part of the Discussion.

4. Methods, study design, lines 346-351. Details on clinical specimen collection and storage should be provided. The authors note that, “children did not receive clinical treatment including antibiotics treatment when samples were collected”. How did the authors account for potential self-medication with non-prescribed antibiotics?

Response: We added detailed information on the sample collection and storage in Methods of the revised version. With respect to the use of antibiotics, because antibiotics belong to prescription medicine, and are banned from being sold without doctors' prescriptions, especially in big cities (e.g. Shanghai), there was very low possibility of self-medication with antibiotics for our study subjects; although we cannot completely rule out that a small number of study subjects might have used antibiotics obtained from irregular sources. .

5. Methods, proteomic chip-based cytokine assay, lines 446-447. Given that acute serum specimens were available for cytokine analysis, did the authors consider adding serologic testing as an indicator of immune status to at least the major respiratory viruses? NGS or PCR analysis of the sera for respiratory viruses/phage might also provide insight into the integrity of airway mucosa.

Response: PCR or qPCR-based assays are the most robust methods to determine the etiological agents of ARTIs from the NP or throat swabs, and have higher sensitivity and detection rate than the serological tests (*J Clin Microbiol.* 2011,

49(9)(suppl):S44-S48). In general, serum samples have very lower detection rate for respiratory viruses than NP swabs. Therefore, PCR-based assays were recommended to detect common respiratory viruses from NP swab samples (*Clin Microbiol Rev.* 2018, 32(1): e00042-18; *JAMA Pediatr.* 2017, 171(8):798-804), and widely used in previous studies (e.g. *J Infect Public Health.* 2018, 11(2):183-186; *Arch Virol.* 2016 Jul; 161(7):1907-13; *Clin Microbiol Infect.* 2012, 18(1):74-80; *J Clin Virol.* 2010, 49(3):211-8; *J Med Virol.* 2014, 86:1249-1255; 2012, 84(12):1980-4; 2012, 84(4):672-8; *PLoS One.* 2013, 8(11):e79477; 2013, 8(8):e72606; 2013, 8(5):e64254; 2012, 7(9):e44568; *Virology J.* 2013, 10:143, etc.).

To further test previous observation, we selected 20 serum samples from children with positive swabs for IFV, HRV, HEV, or RSV in the single group and the multiple group each, and tested the four main respiratory viruses (IFV, HRV, HEV and RSV) using RT-qPCR. Only five serum samples (including three from the single-ARTI group and two in the multiple-ARTIs group) were detected as positive for one of the four viruses (Table R1). Too low detection rate of respiratory viruses among sera by PCR-based assays might not completely reflect the integrity of airway mucosa. Furthermore, the detection of certain respiratory virus among serum might imply a viremia and higher virus titer in the respiratory tract, and is often associated with severe disease conditions and even fatalities (*PLoS One.* 2016, 11(8):e0160777; *Clin Microbiol Rev.* 2018, 32(1): e00042-18). Our results suggest that certain respiratory virus was not significantly associated with the occurrence of multiple ARTIs. Therefore, we did not use the detection of respiratory viruses/phage among sera as a means to reflect the integrity of airway mucosa.

Table R1. Detection of four common respiratory viruses among sera using RT-qPCR assays.

	Single (n=20)	Multiple (n=20)	Total (n=40)
	positive samples	positive samples	
IFV	1	1	2
HRV	0	0	0
HEV	2	1	3
RSV	0	0	0
Total	3	2	5

6. Methods, proteomic chip-based cytokine assay, line 448-449. Were the 20 asymptomatic control children matched for age and gender?

Response: The age and gender of the healthy controls were matched with the ARTI group (Table R2). We provided this information in the revised manuscript.

Table R2. Age and gender information between the ARTI group and healthy control

	The ARTI group	Healthy control	P value
Age (year)	3.5	4.0	0.180
Gender (male/female)	50/59	11/9	0.476

7. Methods, selection criteria ... , lines 366, 369, 410, Table 1. For added clarity, Table 1 should note that 2 of the 50 single ARTI and one of the 62 multiple ARTI children were excluded from the analysis (referral to Table 1 might be added to line 411).

Response: As suggested, we added a note into Table 1 to clarify the sample information, and a referral to Table 1 to the corresponding line.

8. Methods, nucleic extraction ... , lines 437-443. Please clarify the purpose of testing for HRV, HEV, IFV and RSV using specific RT-PCR assays. Were these assays used to determine the “% of samples positive for common RVBs” in Figure 2c?

Response: “Bleed over” is a common problem faced by viral metagenomic analysis (*Nat Commun.* 2018 Oct 15; 9(1):4270), which can cause false positive when a low-cutoff value was used. The *Nat Commun* paper suggested 0.4% as a threshold for “bleed over” to reduce false positives. According to the cutoff used in another virome-related paper (*Nat Med* 2017 Sep; 23 (9):1080-1085) and also the measurement of taxa distribution at different cutoff levels, we chose 15 reads as a cutoff to determine positive or negative results. This cutoff is rigorous enough to reduce false positive results, but possibly lead to false negative results. Because HRV, HEV and IFV were the top three respiratory viruses identified among ARTI children by NGS (Fig. 2a), and none of these viruses was found to be significantly associated with the multiple ARTIs, we used the RT-qPCR assays specific for the three viruses to further confirm the association of certain single virus with the occurrence of multiple ARTIs. The inclusion of RSV was due to the fact that early childhood infection by RSV or rhinovirus had been reported to be associated with recurrent asthma (*New Engl J Med.* 2013; 368: 1398-407; *New Engl J Med.* 2013; 368: 1791-9). The results in Figure 2c were based on the RT-qPCR assays.

With respect to RVBs, we think the reviewer meant are not sure Rhinovirus B (RV-B), if so, the answer is that the RT-qPCR assay for HRV covered Rhinovirus B.

9. Results, Line 117. How do the authors explain their failure to identify human bocavirus, which would be expected to be found in this age group?

Response: We agree that bocavirus is a common respiratory virus causing ARTI in children. However, according to previous molecular epidemiological data by us and other groups (*Arch Virol.* 2016 Jul; 161(7):1907-13; *PLoS One.* 2012;7(9):e44568.) , the prevalence of bocavirus among ARTI children in Shanghai were very low (1.2%

of 2819 children from 2011-2014, and 3.7% of 817 children from 2006-2008). Furthermore, low prevalence of bocavirus was also found in Nanjing (near Shanghai) (1.2%), Wuhan (1.4%) and other cities in China (*J Med Virol* 2012. 84:1980–1984; *J Med Virol* 2012. 84:672–678). Therefore, no detection of bocavirus in this study by the metagenomic analyses might be due to low prevalence and small sample size (total: 212). In addition, because the rigorous metagenomic pipeline was used (please see above), two to three bocavirus-positive cases (1.2% of 212 samples) might have been missed.

10. Results, Line 117 & Fig. S2. The 34 virus species identified should not be co-classified as “pathogenic respiratory viruses”. Although found in the respiratory tract, a suggested clinical classification scheme follows:

1).Respiratory viruses with low proportions of subclinical infections

a. Influenza A & B

b. Human respiratory syncytial virus

c. Human parainfluenza viruses 1-4

d. Human metapneumovirus

e. Human mastadenovirus B

2).Respiratory viruses with high proportions of subclinical infections

a. Rhinovirus A, B, C

b. Enterovirus A, B, D, sp.

c. Influenza C

d. Human mastadenovirus C

e. Human coronaviruses OC43, NL63, 229E

3). Viruses rarely etiologically associated with ARTIs

a. Human herpes viruses

b. Polyomaviruses

4). Viruses not etiologically associated with ARTIs

a. Adeno-associated viruses

b. Papillomaviruses

Response: We thank the reviewer for providing the clinical classification scheme of respiratory viruses, which is very helpful and has been incorporated in the revised manuscript. We changed “34 pathogenic respiratory viruses” as “34 common respiratory viruses” in the revised version. In addition, we added some descriptions to clarify the clinical classification of the respiratory viruses identified by the NGS (e.g. including respiratory viruses with clinical and subclinical infections such as influenza

A & B, C, human respiratory syncytial virus, human parainfluenza viruses 1-4, human metapneumovirus, human mastadenovirus B, rhinovirus A-C, enterovirus A, B, D, human mastadenovirus C, human coronaviruses OC43, NL63, and 229E, as well as viruses rarely associated with ARTIs such as human herpes viruses, polyomaviruses, adeno-associated viruses, papillomaviruses).

11. Results, lines 119-124, Fig 1a. Despite a reasonable \$\geq 30\$ day waiting period between consecutive episodes, single ARTI infections with a definitive respiratory virus pathogen (see above) might trigger downstream reactivation and shedding of DNA viruses that evidence persistent infections and may account for increases in co-detections on subsequent ARTIs. These viruses include herpes viruses, polyomaviruses, papillomaviruses, species C adenoviruses and adeno-associated virus. The authors should consider removing these viruses from their analysis to see if their findings hold.

Response: We thank the reviewer for this very good suggestion. We re-analyzed the results in the Figure 1a-c using the data excluding potential persistent viruses (i.e. herpesviruses, polyomaviruses, papillomaviruses, species C adenoviruses and adeno-associated virus). A consistent trend in the detection/co-detection rates, as well as Shannon diversity index and Chao richness score was observed in the single-ARTI and multiple-ARTIs groups (Figure 1d-f). Because these results of pathogenic respiratory viruses can better reflect the infection conditions of ARTIs, we added these new results into the revised manuscript.

Figure 1d-f in the revised manuscript. Characteristics of respiratory virome among children with single or multiple ARTIs over time.

12. Results, lines 138-139. Can the authors provide a reference showing that bacteriophage “profiles” in the URT correspond quantitatively with their respective bacteria?

Response: Bacteriophages co-evolve with their host bacteria and shape the composition, diversity and function of bacteria community by gene transfer or killing bacteria (*Nat Immunol.* 2013 Jul; 14(7):654-9). We provided related references (*Adv*

Appl Microbiol. 2014, 89:135-83, *Nat Immunol.* 2013, 14(7):654-9; *Cell* 2015, 160, 447–460) to support a correlation between bacteriophages and their host bacteria. The interaction between phages and bacteria is very complex, and in most cases, there is a positive correlation (*Adv Appl Microbiol.* 2014. 89:135-83, *Cell* 2015, 160, 447–460). We did not find literatures to directly support the correlation of bacteriophage profile and their host bacteria in the URT.

On the other hand, to test the correlation between bacteriophage and their host bacteria in URT, we selected seven bacteriophages (including *Propionibacterium* and *Lactococcus* phages) with highest abundance in URT in this study, and performed linear-regression analyses (spearman correlation) using the phage reads and bacteria reads abundance obtained from NGS. We found that most phages (*Propionibacterium* phages, *Lactococcus* phages, *Enterobacter* phages, *Klebsiella* phages, and *Pseudomonas* phages) had a significant but weak positive correlation with their host bacteria (Figure S5 in the Supplementary materials). Other phages (such as *Streptococcus* and *Escherichia* phages) also have a weak or no positive correlation with their host bacteria (*P* value: 0.052 and 0.558). These results suggest that the bacteriophage profile in the URT correspond quantitatively with their host bacteria. We mentioned the results in the revised version (supplementary Figure S5).

Figure S5 in the revised manuscript. The correlations of six top phages with their bacterial hosts. The result of Escherichia phages and their hosts was not shown here.

13. Results, line 156. Do the authors mean “commensal” instead of “communal”?

Response: Yes, we made the suggested change.

14. Results, lines 162-163. Is there a reason that the authors chose *Lactococcus* phage as a comparator with *Propionibacterim* phage from among the available options? It is interesting to note that *Erwinia* phage show a significant but reciprocal difference in levels between single- and multiple-ARTI children and show a progressive decrease with ARTI episodes (Figure S3). By the same logic, is this a potential biomarker for multiple-ARTI?

Response: There are two reasons for choosing *Lactococcus* phages as a comparator of *Propionibacterim* phages. First, *Propionibacterim* and *Lactococcus* phages were the most prevalent and abundant phages identified in the multiple-ARTIs and the single-ARTI groups, respectively (Figure 3b in the revised manuscript). Relative to other phages that had relatively lower prevalent and abundant in the single-ARTI group, we think that the *Lactococcus* phages were more representative for the single-ARTI group than other phages. Furthermore, consistent results were obtained when other phages were used (Figure S4). Second, *Lactococcus* are generally regarded as commensal bacteria in humans (*Front Microbiol.* 2015. 6:20), and some studies reported that *Lactococcus* abundance was decreased in some patients compared to healthy individuals (*Sci Report* 2018. 8:10812; *Appl Environ Microbiol.* 2012. 78(17):6262-70), suggesting an imbalanced microbiota.

Erwinia is a genus of *Enterobacteriaceae* bacteria. *Erwinia* bacteria infect plants and cause disease in plants. They can be isolated from fresh produce and exist in the atmospheric bacterial aerosol (*J Appl Bacteriol.* 1980. 49:175-181). Therefore, it is not surprising that *Erwinia* bacteria (or their phages) are detected in respiratory tract (or NP swabs samples). Like other phages, *Erwinia* phages also have weak positive correlation with their host bacteria (Figure R4). *Erwinia* phages had lower abundance in the multiple-ARTIs group than in the single-ARTI group (Figure 3b and Figure S4), possibly indirectly reflecting a result of the imbalanced microbiota (or dysbiosis) together with other phages like *Lactococcus* and *Enterobacter* phages. Furthermore, the detection rate of *Erwinia* was very low even if in the single-ARTI group. Therefore, we did not think *Erwinia* and other phages (having higher abundance in the single-ARTI group) have diagnostic potential.

Figure R4. The correlations of *Erwiniaphages* with their host bacteria.

15. Discussion, lines 233-235. The authors should be cautious in interpreting their finding of a quantitative correlation of *Propionibacterium* phage levels in the URT with multiple ARTIs. Correlation does not equal causation.

Response: We fully agree with the reviewer's comment. Elevated abundance of

Propionibacterim phages in the multiple-ARTIs group might indirectly reflect the status of changed respiratory microbiota, which might be associated with the susceptibility of the children to respiratory virus infection, rather than represent the causation of frequent ARTIs in children. We have revised the manuscript throughout, and did not draw arbitrary conclusion on the causal relationship between the *Propionibacterium* phages and multiple ARTIs.

16. Discussion, lines 279-290. Can the authors cite any reference that *Propionibacterium* phage or their host bacteria are potentially pathogenic in the URT of otherwise healthy children? If elevated levels of these bacteria in the URT represent a dysbiosis of the URT that could lead to colonization with definitive bacterial pathogens, why not just test for these pathogens directly (*S. pneumoniae*, *H. influenza*, *S. aureus*, *S. pyogenes*, *M. catarrhalis*) using specific PCR assays to evaluate this hypothesis?

Response: *Propionibacterium* is a kind of commensal bacteria in the microbial community of the skin, oral/nasal cavity, and gastrointestinal and genitourinary tracts. It causes infections of the skin, soft tissue, cardiovascular system, or deep-organ tissues (*Clin Microbiol Rev.* 2014. 27(3):419-40). For example, *Propionibacterium acnes* can cause skin wounds (e.g. acne lesions), and was reported to be associated with some diseases (e.g. sciatica) and implant-associated infections (*Lancet.* 2001. 357(9273):2024-5; *Clin Microbiol Rev.* 2014. 27(3):419-40). Furthermore, some studies reported that respiratory tract *Propionibacterium* was associated with ventilator-associated pneumonia, and virus infection (*Arch Oral Biol* 2018. 85. 64–69; *J Allergy Clin Immunol* 2014. 133(4):1220-2; *Microbiome.* 2014. 2:22). *Propionibacterium* can also lead to pulmonary inflammation and is responsible for chronic rhinosinusitis (*Am J Respir Cell Mol Biol*, 2006. 35: 347–356; *BMC Infect Dis.* 2013. 13:210). We cited these papers to support a potential pathogenic role of *Propionibacterium* in respiratory diseases.

Our results suggested that elevated levels of *propionibacterium* phages might indirectly reflect a dysbiosis in the upper respiratory tract, which may further influence the host immunity and lead to a disease status(*Nat Rev Microbiol.* 2017 . 15(5):259-270). However, the dysbiosis of the URT does not automatically lead to higher colonization rates for some specific respiratory bacterial pathogens (e.g. *S. pneumoniae*, *H. influenza*, *S. aureus*, *S. pyogenes* or *M. catarrhalis*). To prove this hypothesis, we selected 42 and 45 available swab samples from the single-ARTI and the multiple-ARTIs groups, respectively, and subjected them to specific PCR assays for *S. pneumoniae*, *H. influenza*, and *S. aureus*. The results showed that there were no significant difference in the detection rates of the three main pathogenic bacteria between the single-ARTI and the multiple-ARTIs groups (Table R3).

Table R3. Detection of *S. pneumoniae*, *H. influenza*, and *S. aureus* by specific PCR assays.

	The single-ARTI group (n=42)	The multiple-ARTIs group (n=45)	Total (n=87)	P value
S. pneumoniae	17 (40.5%)	20 (44.4%)	37 (42.5%)	0.829
H. influenza	32 (76.2%)	34 (75.6%)	66 (75.9%)	1.000
S. aureus	25 (60.0%)	19 (42.2%)	44 (50.6%)	0.135

17. Discussion, lines 273-277. That increased levels of TIMP-1 and PDGF-BB have been found in patients with asthma/COPD, and that these chronic conditions could account for multiple-ARTI episodes (exacerbations) in these study patients, it is essential to ascertain this information as part of the study design.

Response: We reviewed the medical records of the recruited children in this study, and found that only a few children had asthma (2 cases in the single group, 3 cases in the multiple group), and no child had COPD. The low percentage (~2.3%) of asthma in the cohort suggests that asthma or COPD have little influence on the occurrence of multiple ARTIs over time. We added this information in the Discussion section of the revised version of the manuscript.

18. Discussion. Are there practical/cost limitations to using both undirected NGS to assess the URT virome and cytokine analysis of serum as a clinical laboratory test for multiple ARTI susceptibility? Given that a child is identified as potentially susceptible to multiple ARTIs, what practical interventions would the authors suggest that would help reduce this risk?

Response: NGS is a powerful and promising tool for characterizing the microbiota and virome, and investigating their relationship with human diseases. It is also an unbiased tool for pathogen (especially virus) discovery. Similarly, protein array-based cytokine analysis is high-throughput tool to measure cytokine or protein expression profile. These tools are widely used in various fields to find the etiologic agents associated with various human diseases, and look for potential biomarkers for certain diseases. However, we do not encourage to use NGS or array-based cytokine analysis as clinical laboratory tests for clinical diagnosis aim because of the requirement of expensive equipment, high cost, time-consuming analysis and professional operation.

In this study, we found that elevated respiratory *Propionibacterium* phage abundance and serum TIMP-1 and PDGF-BB levels are associated with the occurrence of multiple ARTIs in children using NGS and protein array-based cytokine analyses, which implies diagnostic potential. Given the findings of this study, and

validation by further controlled clinical trials, we think that qPCR and ELISA assays will be ideal choice for clinical use.

With respect to the practical interventions, we recommend improving the immunity of the children by increasing nutrition, maintaining good personal hygiene, avoiding crowded places during high-prevalence of ARTIs or wearing masks when leaving the room, and keeping home environment clean with good indoor ventilation, etc. Furthermore, we suggest more studies on immune intervention strategies.

Reviewers' Comments:

Reviewer #1:

Remarks to the Author:

While the authors did a lot to address the reviewers' comments, some methodological issues remain. These issues limit interpretation of the data and conclusions.

Specific points:

The authors did include age as a covariate, but I don't believe this eliminates the issue that age is a confounder of the interpretation. Age was a significant risk factor for multiple ARTIs based on using the first time point for each individual. It seems that the fact that subsequent ARTIs will occur in children as their ages increase, age is even more of a confounder. It is possible that age-related changes in the microbiome are related to these associations rather than multiple ARTIs.

The authors did not address the issue that use of MDA during sample preparation introduces biases and makes estimation of quantity inaccurate. The authors cite publications in which this was done, but it is still not correct. MDA does induce biases, which affect quantitative measures and Shannon diversity calculations.

See the following papers and others:

Kim et al: APPLIED AND ENVIRONMENTAL MICROBIOLOGY, Nov. 2011, p. 7663–7668

Kim K-H, Chang H-W, Nam Y-D, Roh SW, Kim M-S, Sung Y, et al. Amplification of uncultured single-stranded DNA viruses from rice paddy soil. *Appl Environ Microbiol.* 2008;74:5975–85

Yilmaz S, Allgaier M, Hugenholtz P. Multiple displacement amplification compromises quantitative analysis of metagenomes. *Nat Methods.* 2010;7:943–4. Brinkman, et al *PlosOne*, April 2018, Reducing inherent biases introduced during DNA viral metagenome analyses of municipal wastewater

Furthermore, the authors acknowledge that the samples were not appropriately collected and prepared for bacterial community analysis, but they include correlations with bacteria. If the bacterial data are not solid, as the authors indicate, they should not include bacterial data in the analysis. This could lead to misleading and/or results that are not reproducible.

As I understand the revised description of the analysis pipeline, no translated alignments were done. This is a major limitation of the analysis, as sequences from respiratory viruses are often only captured with translated alignment because of nucleotide sequence divergence from the reference genomes.

The subtraction of 15 reads for "bleed over" seems arbitrary. If using a number of reads rather than a percentage, the depth of sequencing is an important factor. Were the pooled samples/sequencing depth of the samples in this study similar to those in the reference cited? This does not seem like a good way to subtract bleed over because a strong positive is likely to be responsible for more bleed over. That is why percentages of the strong positive are typically used for this type of adjustment.

Reviewer #2:

Remarks to the Author:

Minor comments to Authors:

Line 63, 64. Consider changing "with an estimated annual death of four million" to "with 4 million annual deaths estimated".

Line 124. Consider changing "multiple respiratory virus species may attack the airway over time" to "multiple respiratory virus airway infections over time".

Line 159. Change "rhinovirus" to "HRV".

Line 186. Change "spearman correlation" to "Spearman correlation".

Line 248. Consider changing "demonstrating these three markers" to "demonstrating that together these three markers".

Line 260. Consider changing "that are of medical significance" to "that are of potential medical significance".

Line 366,367. Consider changing "infectious environment would surely protect the child" to "infectious environment might reduce the risk of the child"

Line 403,405. "Significantly, Propionibacteria and TIMP-1 and PDGF-BB could be considered as potential targets for the development of novel therapeutic strategies against the onset of frequent ARTIs in children." These factors are apparently "markers" for multiple ARTIs, but there is no evidence that therapeutic targeting of these markers would have any medical benefit. Perhaps consider rewording.

Line 416, 417. "The diagnostic criteria of ARTI include 1) having a body temperature above 37.5°C; and 2) having at least one of the following symptoms and signs: cough, sore throat, running nose, and expectoration." The unexpected high relative prevalence of influenza viruses compared with RSV in this population might be explained by the fever requirement in the case definition. Infants and young children may not develop fever during acute RSV infections.

Line 433. Change "To identify children experienced" to "To identify children that experienced".

Response to reviewers' comments

Reviewers' comments:

Reviewer #1 (Remarks to the Author):

While the authors did a lot to address the reviewers' comments, some methodological issues remain. These issues limit interpretation of the data and conclusions.

Specific points:

The authors did include age as a covariate, but I don't believe this eliminates the issue that age is a confounder of the interpretation. Age was a significant risk factor for multiple ARTIs based on using the first time point for each individual. It seems that the fact that subsequent ARTIs will occur in children as their ages increase, age is even more of a confounder. It is possible that age-related changes in the microbiome are related to these associations rather than multiple ARTIs.

Response: We agree with the reviewer that age can be a confounder for multiple ARTIs, but its effect is relatively weak in our cohort, as shown with our initial analyses using the first time point for each individual (Table 2 and supplementary Table S2: $p=0.039$). However, age only had OR of 1.5, substantially lower than the OR of 61.5 and 11.6 for the effects of TIMP-1 and PDGF-BB, respectively (Table 2). Importantly, when analyses using the median age of different episodes instead, the age influence was not statistical significant (Table S3-4).

In fact, we emphasized that the children with multiple ARTIs over time might be more susceptible to respiratory virus infection than the children with single ARTIs, as reflected by elevated serum levels of two inflammatory cytokines TIMP-1 and PDGF-BB in the former. We agree that age may be a confounder for subsequent ARTIs as the reviewer has suggested, but we consider age is not a sufficient and necessary condition for multiple ARTIs in our study cohort because subsequent ARTIs were rarely observed in those single-ARTI children during the study period in spite of their age increase (Table 2 and supplementary Tables S2-4).

In contrast, our results highlighted an association between the occurrence of multiple ARTIs in children and an imbalanced respiratory microbiome. As the reviewer has noted that age can influence the composition and profile of microbiome (Gut

Microbiome: What We Do and Don't Know. Nutr Clin Pract. 2015 Dec; 30(6):734-46); but the age-related changes in respiratory microbiome does not necessarily result in imbalanced respiratory microbiome (dysbiosis), and most children will maintain a healthy balanced microbiome as they grow older.

In the revised manuscript, we mentioned that age could be a confounder for multiple ARTIs.

The authors did not address the issue that use of MDA during sample preparation introduces biases and makes estimation of quantity inaccurate. The authors cite publications in which this was done, but it is still not correct. MDA does induce biases, which affect quantitative measures and Shannon diversity calculations.

See the following papers and others:

Kim et al: APPLIED AND ENVIRONMENTAL MICROBIOLOGY, Nov. 2011, p. 7663–7668

Kim K-H, Chang H-W, Nam Y-D, Roh SW, Kim M-S, Sung Y, et al. Amplification of uncultured single-stranded DNA viruses from rice paddy soil. Appl Environ Microbiol. 2008;74:5975–85

Yilmaz S, Allgaier M, Hugenholtz P. Multiple displacement amplification compromises quantitative analysis of metagenomes. Nat Methods. 2010; 7:943–4.

Brinkman, et al PlosOne, April 2018, Reducing inherent biases introduced during DNA viral metagenome analyses of municipal wastewater

Response: As it is the case for any evolving research field, different opinions coexist and not necessarily mutually exclusion. Currently, two main methods, MDA (Qiagen, REPLI-g Single Cell Kit, GE Healthcare illustra GenomiPhi V2 DNA Amplification Kit) and SIA or DOP-PCR (Sigma-Aldrich, GenomePlex Single Cell Whole Genome Amplification Kit), are both widely used for the pre-amplification of viral nucleic acids in virome analysis. Both methods have their advantages and disadvantages. As shown in the papers listed by the reviewer, MDA methods could introduce some biases especially for circular genomes. However, for SIA or DOP-PCR, the nature of exponential amplification can also introduce biases during the amplification step (Annu. Rev. Genomics Hum. Genet. 2015.16:79–102), which may change the real abundance ratio of different copy numbers in the original samples.

The nature of samples sometime dictates the choice of analytical methods.

MDA method is a preferred method for use in the enrichment of low-quantity templates for next generation sequencing (NGS) in many publications (selected list of recent publications: Gut. 2019 Mar 6. pii: gutjnl-2018-318131; Microbiome (2018) 6:68; Proc Natl Acad Sci U S A. 2017, 114(30):E6166-E6175; Cell Host Microbe. 2016, 19 (3):311-22; Emerg Infect Dis. 2015 Jan; 21(1):48-57; Nature. 2010, 466 (7304):334-8). In particular, some studies suggested that MDA method had better performance in NGS coverage, sensitivity and false positive rate than DOP-PCR (Gigascience. 2015, 4:37; Annu. Rev. Genomics Hum. Genet. 2015, 16:79–102).

Therefore, in our study, we used MDA during sample preparation. As we previously explained (*the use of MDA did not specially focus on circular viruses, and the initial large-fragments of DNA templates used for subsequent NGS were not circular. In the analyses, we divided the respiratory virome into three panels, common respiratory viruses, anelloviruses, and bacteriophages. Among bacteriophages, microviridae, a family of bacteriophages that has a single-stranded circular DNA genome, was not preferentially amplified (Figure S2). Therefore, we think that the MDA was less likely to selectively skew our results for circular Propionibacterium phages*), we think that the use of MDA is adequate, and the choice had not biased our conclusions.

In sum, we do respect the reviewer's opinion that the use of MDA may introduce some small biases, and therefore discussed this issue in the limitation of our study and cited all references suggested by the reviewer in the revised manuscript.

Furthermore, the authors acknowledge that the samples were not appropriately collected and prepared for bacterial community analysis, but they include correlations with bacteria. If the bacterial data are not solid, as the authors indicate, they should not include bacterial data in the analysis. This could lead to misleading and/or results that are not reproducible.

Response: We agree with the reviewer that the bacterial data are not solid because the experiment pipeline was not designed for bacterial community analysis. To avoid misleading, we removed this result from the revised manuscript.

As I understand the revised description of the analysis pipeline, no translated alignments were done. This is a major limitation of the analysis, as sequences from respiratory viruses are often only captured with translated alignment because of nucleotide sequence divergence from the reference genomes.

Response: We agree with the reviewer that using translated alignment can capture more unclassified respiratory viruses, and is a robust tool to find and identify novel and divergent viruses. However, our study was not designed for the discovery of highly divergent, novel human viruses, which requires the use of blastx with translated alignment and other less stringent similarity strategy. One of the main goals of this study was to find the potential association of common respiratory viruses with multiple ARTIs. The same strategy used in our study (nucleic acid (NA) based method) can be found in some other studies (PLoS Pathog, 2017, 13(3):e1006292; Nat Med 2017, 23 (9):1080-1085).

We found higher respiratory virus infection rate among children with multiple ARTIs over time than among those with single ARTI. We believed that novel unclassified respiratory viruses were rare in our samples and had less influence on our results. To prove this, we performed blastx-searches using unmapped sequences from previous analyses against viral ref. nr and NCBI nr database. Then, we combined the results of both blastx-searches and nucleic acid (NA) based method, and compared it with our previous results. We found that most of the newly detected viruses belong to unclassified viruses. Although slightly higher detection rates of respiratory viruses were observed in each group by the combined strategy of blastx- and NA based method than the NA based method (i.e. previous results), the results (higher detection and co-detection rates among multiple-ARTI groups) were not significantly changed (Figure R1 vs. Figure 1a in Text).

Similarly, the detection rates of most respiratory viruses were not changed, except for slightly higher detection rates of rhinovirus and enterovirus by the combined strategy of blastx- and the NA based method (Figure R2 vs. Figure 2a). Rhinovirus and enterovirus are highly diverse RNA viruses with higher genetic diversity, and most unclassified respiratory viruses belonged to these two viruses (Figure R2). Because the phylogeny, pathogenesis and immunology of these unclassified respiratory virus infection are unclear, we did not add the unclassified viruses in the revised

manuscript.

Figure 1a in the Text.

Figure R1. Combined results of both blastx and NA based method.

Figure 2a in the Text. Detection rates of common respiratory viruses.

Figure R2. Combined results of both blastx and NA based method.

Moreover, we also analyzed the profile of anelloviruses. Similar to the results of respiratory viruses, inclusion of Blastx did not significantly change the results (Figure R3 vs. Figure 1g, and Figure R4 vs. Figure 2b). In particular, except for the identification of some unclassified anelloviruses, almost all the annotated viruses remained the same detection rate as before (Figure R4 vs. Figure 2b). Furthermore, even though several other anellovirus species were detected, they only appeared 1- 3

of 212 samples (Figure R5).

Figure 1g in the Text.

Figure R3. Combined results of both blastx and NA based method.

Figure 2b in the Text. Detection rates of anelloviruses.

Figure R4. Combined results of both blastx and NA based method. The arrow highlights the unclassified anelloviruses.

Figure R5. Several anellovirus species identified by both blastx and NA based method. Other rare anelloviruses detected in only a few samples (n<3).

The subtraction of 15 reads for “bleed over” seems arbitrary. If using a number of reads rather than a percentage, the depth of sequencing is an important factor. Were the pooled samples/sequencing depth of the samples in this study similar to those in the reference cited? This does not seem like a good way to subtract bleed over because a strong positive is likely to be responsible for more bleed over. That is why percentages of the strong positive are typically used for this type of adjustment.

Response: Thank you for your comments and suggestions. In our previous response, we explained the reason and rationality of using 15 reads as a cutoff to avoid false positive (we showed again below). In fact, we selected a similar pipeline to the Nat Med paper (2017, 23 (9):1080-1085). Neither the Nat Med paper nor our study used pooled samples. Comparison showed that the two studies had similar sequencing depth (Nat Med: median of 29 million reads/sample; this study: 37 million reads/sample) and percentage of overall viral reads (Nat Med: 0.22% (IQR, 0.001 - 1.6%); this study: 0.23% (0.005% - 0.026%)) (Table S6). In particular, sequencing depths were similar among samples (Table S6).

Some researchers suggest that a threshold should be used to exclude potential false positives for samples with similar sequencing depths (Nat Commun. 2018, 9(1):4270.). In this respect, thresholds using a percentage and a reads number should result in similar conclusions for samples with similar sequencing depth. We did not use 0.4% threshold described in a Nat Commun paper (The reason is that they used a Miseq sequencing platform to generate relative longer read length (Miseq) and lower depth (750,000 reads/sample), whereas our study and the Nat Med paper obtained shorter read length (100-150 nt by BGISEQ500 and HiSeq2500) and higher depth (37

and 29 million reads/sample, respectively). Therefore, we hold fast on our stand in this issue as we have responded during the first round of revision, as follows:

Previous response: *“False positive caused by “bleed over” is a common problem faced by viral metagenomic analysis (Nat Commun. 2018, 9(1):4270). In the Nat Commun paper, the authors used 0.4% as a threshold for “bleed over” to reduce false positives. In another study of human virome (Nat Med. 2017, 23 (9):1080-1085), 15 reads were used as the cut off for a positive result. In our study we chose a cut off of 15 reads, which is same to the cutoff used in the Nat Med paper. Furthermore, we measured the taxa distribution at different cutoff levels from 1 to 50 (Figure R1). We found that the number of taxa identified remained stable when the cutoff were ≥ 15 reads (Figure R1). Therefore, we believed that the cutoff of 15 reads was more rigorous to avoid false positive results. Furthermore, for the top eukaryotic viruses identified in this study, further confirmation was also performed by qPCR (Figure 2). The related description and references for the choice of cutoff were added in Methods of the revised manuscript.*

One of the most important findings in this study was that higher abundance of Propionibacterium phages in children with multiple ARTIs over time than in the single-ARTI children, and the reverse for Lactococcus phages and other phages. The cutoff influences some of the judgments of positive and negative, but does not change the relative abundance of each phage in the single-ARTI and the multiple-ARTIs groups. To exclude the potential influence of the cutoff value on the results, we performed similar analyses to Figure 3b without a cutoff (i.e. a cutoff of 0, which means one or more reads representing a positive result). We found that although there was a little change in the detection rates of some phages, the distribution trends of the top 15 phages between the single-ARTI and multiple-ARTIs groups were completely consistent with those determined with a cutoff of 15 reads (Figure 3b and Figure R2).

In addition, we also re-performed the analyses presented in the Figure 4c-f. The results showed that TIMP-1 and PDGF-BB levels are still higher in Propionibacterium phages-positive group than in the negative group when no cutoff was used (Figure R3). These results indicated that the cutoff levels did not influence the main results and conclusions of this study.”

Figure R1 in previous rebuttal. Taxa identified with different cutoff of reads (different colors represent different sequencing batch).

Figure R2 in previous rebuttal. Comparison of the relative abundance of each bacteriophage taxa between children with single and multiple ARTIs using Mann-Whitney U test.

Figure R3 in previous rebuttal. Comparison of TIMP-1 and PDGF-BB levels between children who were positive and negative for Propionibacterium phage was made by using Mann-Whitney U test.

Reviewer #2 (Remarks to the Author):

Minor comments to Authors:

1. Line 63, 64. Consider changing “with an estimated annual death of four million” to “with 4 million annual deaths estimated”.
2. Line 124. Consider changing “multiple respiratory virus species may attack the airway over time” to “multiple respiratory virus airway infections over time”.
3. Line 159. Change “rhinovirus” to “HRV”.
4. Line 186. Change “spearman correlation” to “Spearman correlation”.
5. Line 248. Consider changing “demonstrating these three markers” to “demonstrating that together these three markers”.
6. Line 260. Consider changing “that are of medical significance” to “that are of potential medical significance”.
7. Line 366,367. Consider changing “infectious environment would surely protect

the child” to “infectious environment might reduce the risk of the child”.

Response to comments 1-7: We appreciate the reviewer for kindly editing our manuscript and their help for improving the readability of our paper. All suggested changes were made accordingly.

8. Line 403,405. “Significantly, Propionibacteria and TIMP-1 and PDGF-BB could be considered as potential targets for the development of novel therapeutic strategies against the onset of frequent ARTIs in children.” These factors are apparently “markers” for multiple ARTIs, but there is no evidence that therapeutic targeting of these markers would have any medical benefit. Perhaps consider rewording.

Response: We agree with the reviewer, and removed this sentence from the revised manuscript.

9. Line 416, 417. “The diagnostic criteria of ARTI include 1) having a body temperature above 37.5°C; and 2) having at least one of the following symptoms and signs: cough, sore throat, running nose, and expectoration.” The unexpected high relative prevalence of influenza viruses compared with RSV in this population might be explained by the fever requirement in the case definition. Infants and young children may not develop fever during acute RSV infections.

Response: We agree that relatively high prevalence of influenza viruses compared with RSV in this population might be caused by the fever requirement in the case definition. But our results suggested that respiratory virus (including influenza virus or RSV) itself might not be associated with multiple ARTIs in children over time, so we did not discuss it further.

10. Line 433. Change “To identify children experienced” to “To identify children that experienced”.

Response: We appreciate the reviewer for editing.